

# Effect of elevation feedbacks and climate mitigation on future Greenland ice sheet melt

Thirza Feenstra[1], Miren Vizcaino[1], Bert Wouters[1], Michele Petrini[2], Raymond Sellevold[3], and Katherine Thayer-Calder[4]

[1]Geoscience and Remote Sensing, Delft University of Technology, Delft, The Netherlands
[2]NORCE Norwegian Research Centre AS, Bjerknes Centre for Climate Research, Bergen, Norway, Associate to the National Institute of Oceanography and Applied Geophysics (OGS), Trieste, Italy
[3]Å Energi Fornybar Forvaltning AS, Kristiansand, Norway
[4]Climate and Global Dynamics Laboratory, National Center for Atmospheric Research (NCAR), Boulder, CO, USA

**Correspondence:** Thirza Feenstra (thirzafeenstra@outlook.com)

**Abstract.** The Greenland Ice Sheet (GrIS) stores freshwater equal to more than seven meters of potential sea level rise and strongly interacts with the Arctic, North Atlantic and global climate. Over the last decades, the GrIS has been losing mass and is projected to lose mass at an increasing rate. Interactions between the GrIS and the climate have the potential to amplify or reduce GrIS mass balance responses to ongoing and projected warming. Here, we investigate the impact of ice sheet-climate interactions on the climate and mass balance of the GrIS using the Community Ice Sheet Model version 2 coupled to the

Community Earth System Model version 2 (CESM2-CISM2). To this end, we compare two idealized simulations with a non-evolving and evolving ice sheet topography in which we apply an annual 1% increase in $CO_2$ concentrations until stabilization at four times pre-industrial (PI) $CO_2$ concentrations ($4xCO_2$). By comparing the 1- and 2-way coupled simulations, we find significant changes in atmospheric blocking, precipitation and cloud formation over Greenland as the GrIS topography evolves,

acting as negative feedbacks on mass loss. We also find that a uniform temperature lapse rate misrepresents temperature changes in the ablation area, leading to an overestimation of the positive melt-elevation feedback in the 1-way coupled simulation, resulting in an overestimation of mass loss. Furthermore, we analyze an idealized simulation in which we first apply a $4xPI\ CO_2$ forcing and thereafter annually reduce atmospheric $CO_2$ by 5% until PI concentrations are reached. During the 350 year $4xCO_2$ forcing period, the ice sheet loses a total mass of 1.1 m sea level equivalent, and part of its margins retreat landward. When the

PI $CO_2$ concentration is restored, melt decreases rapidly, leading to a small positive surface mass balance. Combined with the strongly reduced ice discharge resulting from the widespread retreat of the ice sheet margin, this results in the halting of GrIS mass loss, despite a remaining global warming of 2 K. The GrIS, Arctic and North Atlantic ocean strongly interact, causing a complex transitional phase towards a colder climate during the century following the $CO_2$ reduction. Elevated atmospheric temperatures, larger ocean heat transport and a deteriorated state of the snowpack, compared to the initial pre-industrial state,

result in limited regrowth of the ice sheet under reintroduced PI $CO_2$ conditions.





# 1 Introduction

Over the past century, the rate of global mean sea level rise (SLR) has exceeded any previous period (Fox-Kemper et al., 2021) and has been accelerating since the late 1960s (Dangendorf et al., 2019). Melt of the Greenland ice sheet (GrIS), storing more than seven meters of potential SLR (Morlighem et al., 2017), has caused ∼18% of contemporary SLR (Otosaka et al., 2023) and its contribution is expected to increase (Bamber et al., 2019; Goelzer et al., 2020). The Sixth Assessment Report (AR6) of the Intergovernmental Panel on Climate Change (IPCC; Fox-Kemper et al., 2021) attributes the largest uncertainties in future sea level rise to the GrIS and the Antarctic ice sheet. Therefore, understanding the physical drivers for GrIS mass loss, and adequate numerical modeling of their effects on the GrIS mass balance is crucial to provide reliable projections of global and regional SLR.

Contemporary climate change, caused by anthropogenic greenhouse gas emissions (Jones et al., 2013; Ribes and Terray, 2013), influences surface mass processes over the GrIS in connection with rising Northern Hemisphere summer temperatures (Hanna et al., 2008). Moreover, changes in the GrIS have the potential to influence local and global climate (Vizcaíno, 2014) because of its interaction with key components of the Earth system through various processes and feedbacks (Fyke et al., 2018). Important feedback mechanisms include the positive melt-elevation feedback (Edwards et al., 2014; Vizcaíno, 2014; Fyke et al., 2018), the positive melt-albedo feedback (Box et al., 2012; Fyke et al., 2018) and the negative melt-discharge feedback (Goelzer et al., 2013; Fürst et al., 2015; Vizcaíno et al., 2015). The strong coupling between the GrIS and the atmosphere potentially influences atmospheric circulation and precipitation patterns (Ridley et al., 2005; Fyke et al., 2018). The GrIS heavily interacts with the ocean, as the North Atlantic Meridional Overturning Circulation (NAMOC) influences GrIS temperatures, while freshwater influxes as a result of GrIS mass loss can influence the strength of the NAMOC (Driesschaert et al., 2007). Accounting for these interactions between the Earth system and the GrIS in climate models is critical to obtain reliable projections of SLR and GrIS evolution (Vizcaíno, 2014; Fyke et al., 2018). It has been shown that 1-way coupled simulations, in which changes in the state of the GrIS are not communicated to the Earth system model, overestimate multi-centennial GrIS mass loss (Ridley et al., 2005; Gregory et al., 2020), as several feedbacks are not represented in these simulations. Ridley et al. (2005) identify more precipitation and less melt in a 2-way coupled simulation, in which the GrIS is an interactive component within the Earth system model, and attribute this to a change in atmospheric circulation patterns over Greenland. In contrast, Gregory et al. (2020) attribute the reduced mass loss to an increase in cloud fraction, causing an increase in reflected shortwave radiation. Next to that, they find a land inward migration of precipitation patterns caused by topographic changes.

Considering that by 2022, global temperatures have increased by 1.09°C (0.95 - 1.20) compared to the 1850 to 1900 baseline (Gulev et al., 2023), reaching the 1.5°C goal of the Paris Agreement is becoming less likely over time (Matthews and Wynes, 2022). The rapidly increasing global temperatures call for the investigation of 'overshoot' scenarios. Applying a temperature overshoot to the GrIS could have large implications for the evolution of the GrIS-induced SLR, as GMSL could rise substantially under the larger temperatures during such a period (Schwinger et al., 2022). Due to the nonlinear and tipping nature of the Earth system, there might be non-reversible effects associated with an overshoot scenario. Consequently, investigating the response of the GrIS to these kinds of scenarios is of great importance to assess the feasibility and implications of following an





overshoot pathway. Previous work (Ridley et al., 2010; Robinson et al., 2012; Gregory et al., 2020; Bochow et al., 2023) shows that temperature and volume thresholds for irreversible GrIS mass loss might exist. If temperature overshoots are limited, and the ice sheet does not lose a critical ice volume, mass loss can be halted or might even be reversible. However, not all models account for ice sheet-climate interactions and an assessment of the interaction between the evolution of the climate and the surface mass balance in determining whether or not deglaciation rates can be reversed is lacking. As ice sheet-climate interac-

tions could potentially accelerate or slow down the changes in GMSL caused by the response of the GrIS to $CO_2$ reduction, it is important to account for them to obtain a reliable projection of SLR in an overshoot scenario.

With the development of a coupling (Muntjewerf et al., 2021) between the Community Earth System Model version 2 (CESM2; Danabasoglu et al., 2020) and the Community Ice Sheet Model version 2 (CISM2; Lipscomb et al., 2019), it is possible to account for the ice sheet-climate interactions resulting from a dynamically evolving ice sheet. CESM2 can produce

a realistic surface mass balance (SMB; van Kampenhout et al., 2020) by using a surface energy balance (SEB) scheme and accounts for the interactions between the different model components. Several studies (e.g. Muntjewerf et al., 2020a, b; Sommers et al., 2021) have analyzed coupled CESM2-CISM2 simulations, but the effects of incorporating this coupling have not yet been quantified.

In this paper, we use the coupled configuration of CESM2-CISM2 to assess the impact of ice sheet-climate interactions

on the mass balance of the GrIS. We evaluate the impact of these interactions under an idealized extreme warming scenario, in which we annually increase atmospheric $CO_2$ by 1% from pre-industrial (PI) concentrations until four times PI $CO_2$. We begin by comparing two simulations with this idealized forcing, of which one allows changes in the GrIS topography to be communicated to the Earth system model (2-way coupling) and one does not (1-way coupling). We then branch a third simulation from the 2-way coupled simulation at year 350 (i.e. after two centuries of sustained 4x$CO_2$ conditions), where we

bring back PI $CO_2$ concentrations with an annual 5% decrease (covering 27 years), to assess the impact of ice sheet-climate interactions on the response of GrIS deglaciation rates to $CO_2$ reduction.

In section 2 we describe the CESM2-CISM2 model setup, the simulations conducted and our analysis methods. Section 3 assesses the influence of the coupling on the simulated GrIS mass loss, and section 4 gives an overview of the most important climate-ice sheet interactions that influence the GrIS mass balance. In section 5 we address the GrIS mass balance response to

the reintroduction of PI $CO_2$ conditions and investigate the related climate-ice sheet interactions. We discuss our results and draw conclusions in sections 6 and 7 respectively.

## 2 Method

### 2.1 Model description

We use the Community Earth System Model version 2 (CESM2; Danabasoglu et al., 2020) and the Community Ice Sheet

Model version 2 (CISM2; Lipscomb et al., 2019), which are coupled to account for an evolving ice sheet. CESM2 is a state-of-the-art community-developed Earth System Model (ESM), developed by the U.S. National Science Foundation (NSF) National Center for Atmospheric Research (NCAR). CESM2 consists of different component models for land and land biochemistry,





atmosphere, river runoff, surface waves, ocean and marine biochemistry, sea ice and land ice, which are coupled to exchange states and fluxes via a hub and spoke architecture (Danabasoglu et al., 2020).

The atmosphere model is the Community Atmosphere Model version 6 (CAM6; Gettelman et al., 2019), which runs on a nominal 1° (1.25° in longitude and 0.9° in latitude) grid and has 32 vertical levels. The Community Land Model version 5 (CLM5; Lawrence et al., 2019) shares the CAM6 horizontal grid. In CLM5, every grid cell is divided into one or multiple fractions of land units, which can be glacier, lake, wetland, urban, vegetation and crop surface. Calculations are carried out separately over the different land units. The model has a fixed number of vertical layers for the soil, whereas there is a variable

number of layers for snow and firn, with a maximum snow depth of 10 m water equivalent (w.e.). The model allows for compaction of snow into firn. Accumulation of snow over 10 m w.e. in a grid cell is transferred as positive SMB to CISM2. If snow and firn are melted away, further melt is transferred as negative SMB (ice ablation) to CISM2. The snow albedo is calculated with the SNow, ICe and Aerosol Radiation Model (SNICAR; Flanner et al., 2021). The surface runoff from melt and rain is routed to the ocean using the Model for Scale Adaptive River Transport (MOSART; Li et al., 2013). Modeling the

ocean is done using the Parallel Ocean Program version 2 (POP2; Smith et al., 2010). POP2 has a resolution of a nominal 1°, with a uniform resolution equal to 1.125° in the zonal direction. The sea ice model is the Los Alamos National Laboratory sea ice model version 5 (CICE5; Hunke et al., 2017). CICE5 shares the same horizontal grid as POP2. The model consists of an ice thermodynamic model, an ice dynamics model and a transport model that computes both horizontal transport as well as transport in thickness space.

The GrIS is modeled using the Community Ice Sheet Model version 2 (CISM2; Lipscomb et al., 2019). CISM2 is a parallel 3D thermodynamic model, which solves the momentum balance and computes the ice sheet thickness and temperature (Lipscomb et al., 2019). CISM2 has a 4 km rectangular grid, with 11 vertical sigma levels. The model uses approximations of the Stokes equations for incompressible viscous flow and a pseudo-plastic sliding law to parameterize basal sliding. Floating ice at marine margins is immediately discharged to the ocean using a flotation criterion (Muntjewerf et al., 2021).

## 2.2 Coupling description

By coupling CISM2 with CESM2, interactions and feedbacks between the ice sheet and the climate are accounted for in the projected evolution of GrIS mass loss. The coupling of CISM2 is bi-directional with CLM5 and CAM6 and uni-directional with POP2 (Figure 1) and has an annual frequency, except for the communication of the GrIS topography to CAM6, which is done every 5 or 10 model years. In CLM5, the SMB is computed by subtracting the runoff and sublimation from the precipitation

over the ice sheet. Precipitation is the sum of snow and rain. Snow directly contributes positively to the SMB, while rain can either have a net SMB contribution of zero if it runs off or a positive contribution if it refreezes. Melted snow and ice can also be divided into a fraction that contributes to the runoff and a fraction that refreezes. Therefore, additionally, we can express the SMB as the sum of melt and sublimation subtracted from the sum of snow and refreezing. Then, we define the refreezing capacity as the amount of refreezing divided by the amount of available water, which consists of meltwater and

rainfall. The model computes the energy available for melt using the surface energy balance (SEB). The melt energy is equal to the sum of the net radiative heat flux (the sum of shortwave and longwave radiation), the turbulent heat flux (the sum of





latent and sensible heat) and the ground heat flux. The SMB and SEB are computed for multiple elevation classes (Sellevold et al., 2019), which allows for resolving the large heterogeneity around the steep ice sheet margins and for remapping of the SMB on the ice sheet model grid, which is done using trilinear interpolation and renormalization (Muntjewerf et al., 2021).

For every elevation class, the surface energy fluxes from CLM5 are scaled down, after which the SMB is computed. The grid cell temperature is downscaled using a uniform lapse rate of -6 K km$^{-1}$. Precipitation falls as snow if the elevation-corrected near-surface temperature is below -2°C and as rain if this temperature is above 0°C. Between -2°C and 0°C precipitation falls as a mix of rain and snow (Muntjewerf et al., 2021). Then, in CISM2, the total GrIS mass balance (MB) is computed as the sum of the downscaled SMB, basal mass balance and ice discharge. As the ice sheet loses or gains mass, the updated topography

and ice sheet margins are communicated to CLM5. In CLM5, the land units and topographic height are updated according to the updated ice sheet margins and topography, redistributing the weights of the different elevation classes (Muntjewerf et al., 2021). The changing ice sheet topography does not only influence the computations in CLM5 but those in CAM6 as well, as the topography influences atmospheric circulation. Therefore the ice sheet topography computed in CISM2 is communicated to CAM6, where it is remapped to the CAM6 grid in the form of surface geopotential height (Muntjewerf et al., 2021).

Regarding the uni-directional coupling with the ocean, CISM2 computes the annual ice discharge and basal melting, which is communicated to POP2 and then supplied to the ocean at a constant rate throughout the following year. Coupling from POP2 to CISM2 is not yet implemented in CESM2. Therefore ocean-forced melting of marine-terminating glaciers is not accounted for (Muntjewerf et al., 2021). Runoff as a result of melt and rain is computed in CLM5 and is distributed to the ocean using MOSART and is therefore seasonally varying.

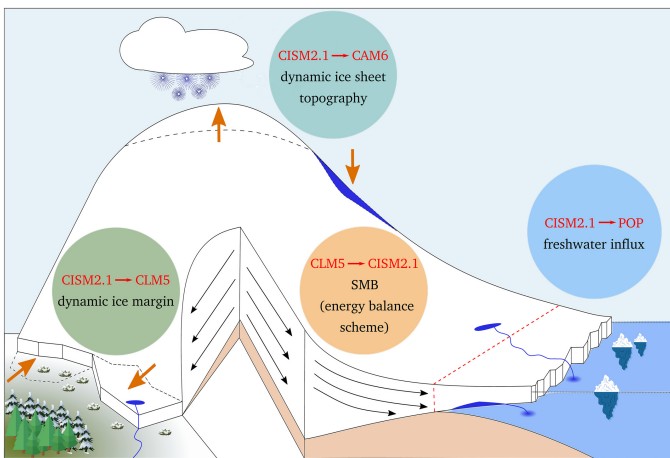

**Figure 1.** Schematic of four elements of coupling between ice sheets and other Earth system components, courtesy of M. Petrini (Muntjewerf et al., 2021). The land model (CLM5) receives the location of the ice sheet margin from the ice sheet model (CISM2). The land and the atmosphere (CAM6) models receive the dynamic GrIS topography from CISM2. CISM2 provides the freshwater influx from ice discharge to the ocean model (POP), while CLM5 provides the freshwater fluxes from surface runoff. The ice sheet model itself receives the SMB from CLM5 to compute changes in GrIS topography and extent.





## 2.3 Simulation design

In this study, we analyze both 1- and 2-way coupled simulations. The 1-way coupled simulation is a simulation in which changes in GrIS elevation and extent are not communicated from CISM2 to the other model components. The GrIS topography is constant in the land and the atmosphere model and the freshwater fluxes from the GrIS to the ocean model are fixed. The topography and freshwater fluxes in the 1-way configuration are the same as the initial state of the 2-way coupled simulation. The resulting climate and SMB are downscaled using elevation classes (Sellevold et al., 2019), using a temperature lapse rate of -6 K/km and are interpolated onto the CISM grid, in the same way as is done in the 2-way coupled configuration. The coupler communicates the downscaled SMB to CISM2, where a new ice sheet topography is computed. However, this is not communicated to the other model components when using 1-way coupling. In contrast to the 1-way coupled run, the 2-way coupled run communicates the changes in the GrIS topography and ice sheet processes computed in CISM2 to the other model components every model year. This means that POP2 will receive updated freshwater fluxes from ice discharge and that CLM5 and CAM6 will receive an updated topography from CISM2. Therefore, topography-related feedbacks will affect the state of the GrIS and the climate. In contrast to the surface topography, the surface albedo is updated as a response to ice sheet melting in both the 1-way and 2-way coupled simulations.

We begin with a comparison of a 1-way and a 2-way coupled simulation both forced with a 4x$CO_2$ scenario. In this scenario, we apply a yearly 1% increase in $CO_2$ concentration until four times the PI $CO_2$ concentration (1140 ppm) is reached in year 140. After year 140, we keep the $CO_2$ concentration constant at 4x$CO_2$. The 4x$CO_2$ scenario is an extreme warming scenario, which after the year 140 has a similar radiative forcing as the SSP5-85 scenario by the end of the twenty-first century (Muntjewerf et al., 2020a). Next, we analyze another simulation branched from our 2-way coupled simulation, in which we decrease the $CO_2$ concentration by an annual 5% between years 350 and 377 until PI $CO_2$ conditions are reached, after which the $CO_2$ concentration remains constant. We analyze 500 simulation years when comparing the 1- and 2-way coupled simulation and 925 simulation years of the $CO_2$ reduction simulation. The climate and ice sheet exchange information every year for the first 500 years of the simulation. After year 500, the climate does not change rapidly anymore. Therefore, to save computational resources, the coupling is done every 5 years after the year 500. Next to the $CO_2$ forcing experiments, we use a 300 year 2-way coupled PI control simulation (Danabasoglu, 2019) with a constant PI $CO_2$ concentration for comparison.

## 2.4 Definition of ice sheet and climate metrics

Besides considering the mass and energy balances computed in CISM2 and CLM5, we use several other ice sheet metrics. We compute the equilibrium line altitude (ELA) from the hypsometric curve, which represents the cumulative area distribution of the ice sheet with respect to elevation. The mean ELA is then defined as the elevation corresponding to the extent of the ablation area, which is the area with a negative SMB. We consider lapse rates to assess the effect of elevation on ice sheet surface processes. The lapse rates are computed by dividing the difference in the mean state of a variable between the 1- and 2-way simulation by the difference in the elevation between the 1- and 2-way simulation for the years 480-500. Thresholds of 250 m elevation difference and an ice sheet fraction of 90% of the grid cell are taken to exclude grid cells in which most of the





changes are not caused by elevation changes. It should be noted that these lapse rates describe the change in a variable resulting from GrIS surface changes, rather than the rate of change over a changing pressure level as described by the free atmospheric lapse rate.

We assess changes in atmospheric circulation by considering Greenland blocking events, the North Atlantic Oscillation (NAO) and integrated vapor transport. The Greenland Blocking Index (GBI) is computed using the method proposed by Hanna et al. (2018), for both the summer (JJA) and winter (DJF). We subtract the area-weighted mean of the 500 hPa geopotential height over the Arctic region (60–80° N) from the Greenland region (60–80° N, 20–80° W) and normalize with respect to the control simulation. We define the NAO index as the principle component corresponding to the leading empirical orthogonal function (EOF) of the seasonal (JJA and DJF) sea-level pressure in the North Atlantic region (20-80° N, 90° W - 40° E), normalized with respect to the control simulation (Hurrell, 1995). The integrated vapor transport (IVT) north- and eastward components are computed following Reynolds et al. (2022), using:

$$\text{IVT} = -\frac{1}{g} \int\limits_{1000}^{300} qV \, dp$$

where $g$ is gravity, $q$ is specific humidity, $V$ is wind velocity and $p$ is pressure. We integrate from 1000 to 300 hPa. We compute the total IVT from the north- and eastward IVT components.

Finally, the North Atlantic Meridional Overturning Circulation (NAMOC) index is defined as the maximum strength of the overturning stream function north of 28° N and below 500 m depth and the Arctic sea ice extent is defined as the area north of 60° N where sea ice concentration is greater than 15%.

We consider 20 data points in time for centered moving averages and periodic means to obtain a climatological mean state and variability. This means that before the year 500, 20 years are considered, whereas after the year 500, 100 years are considered. As the climate is evolving less rapidly after the year 500, a 100 year mean will be able to represent the mean state of the climate, while being consistent in terms of variability compared to the period before the year 500.

## 2.5 Emergence and recovery

We use the emergence concept applied by Fyke et al. (2014) to assess whether the differences between the 1- and 2-way simulations are significant. We define the first year of significant difference as the first year that the 20 year centered moving average of the differences between the 1- and 2-way simulation has emerged from the natural variability of the differences. We define the natural variability of the differences as all absolute values smaller than one standard deviation of the differences ($1\sigma_{\text{differences}}$). To obtain $\sigma_{\text{differences}}$, we use the standard deviations computed from the 2-way coupled PI control simulation ($\sigma_{\text{control}}$). We assume that this control simulation can represent the mean state of a similar 1-way control simulation, as the ice sheet is nearly in equilibrium, having a limited mean SLR rate of 0.03 mm yr$^{-1}$, and its variance only represents natural variability. Then, by using error propagation, we can express the $1\sigma_{\text{differences}}$ interval in terms of $\sigma_{\text{control}}$ as [-$\sigma_{\text{control}}\sqrt{2}$, $\sigma_{\text{control}}\sqrt{2}$]. When the 20 year centered moving average of the difference has migrated permanently outside the $1\sigma$ interval, we consider the signal emerged.





To assess whether climate and ice sheet variables can recover to their PI state after the reintroduction of PI $CO_2$ levels, we
apply a modified approach of the aforementioned emergence concept. A variable is considered recovered if the 20 year centered
moving average returns within one standard deviation ($1\sigma$) from the PI mean and stays within the $1\sigma$ interval (obtained from the
PI control simulation) for the remainder of the simulation. In case a variable has not recovered by the end of the simulation, we
compute an extrapolated year of recovery from the remaining linear trend over the last 100 years (825 - 925) and the remaining
anomaly of the mean state of the variable in the last 100 years of the simulation with respect to the closest boundary of the $1\sigma$
confidence interval. We compute the relative amount of recovery, by comparing the remaining anomaly in the final 100 years
of the simulation to the maximum response during the 4x$CO_2$ forcing period.

## 3    Simulated mass loss in the 1- and 2-way coupled simulation

To assess the impact of simulating an interactive ice sheet, we first consider differences in the global and regional climate, to
see whether this affects or is affected by GrIS mass loss. As a response to the $CO_2$ forcing (Figure 2a), global temperatures
rise, with a global warming response of nearly 10 K by year 500, which is not affected by the coupling (Figure 2b). Under
the 4x$CO_2$ forcing, the North Atlantic Meridional Overturning Circulation (NAMOC) collapses in around 150 years in both
simulations (Figure 2c). A near-zero GrIS integrated mass balance in the first 120 years (Figure 2d) results in limited mass
loss in this period (Figure 2e). During these years, the NAMOC evolution in the 1- and 2-way coupled simulations is similar,
showing that this NAMOC collapse is not strongly related to increased freshwater fluxes from the GrIS.

After year 120, there is an accelerating decrease in SMB, which is strongest in the 1-way coupled simulation (Figure 2d,
Figure A1g-j), resulting in the separation of the SLR curves at the year 282 (Figure 2e). In contrast, the contribution from ice
discharge decreases in both simulations due to the melt-discharge feedback, as the ice sheet starts retreating (Figure A1b-e).
Because of the small ice discharge contribution in both simulations, the total mass balance differences emerge in the same
225    year as the SMB. After emergence, the 1-way coupled simulation consistently projects $17.0 \pm 0.4\%$ more mass loss ($p < 0.01$).
GrIS area loss acceleration is slightly delayed compared to the mass loss, and the differences become significant in the year 300
(Figure 2f). Ice mass is mainly lost at the margins, although less strongly in the 2-way coupled simulation (Figure A1b-e). The
increased melting of the ice sheet is reflected in the rise of the equilibrium line altitude (ELA; Figure 2g) and the expansion of
the ablation area (the area with negative SMB; Figure 2h). As melting is stronger in the 1-way case, the ablation area expands
faster and the ELA heightens faster. However, the ELA and extent of the ablation area of the two simulations converge in years
320 and 400 respectively. As the GrIS loses mass, the equilibrium line moves more inland, but its altitude does not increase
strongly anymore, as the ice sheet's elevation decreases simultaneously. As the ice sheet extent becomes smaller, parts of the
ablation area are lost, slowing down the increase in the total percentage of ablation area. The period of this slow increase is
reached later in the 2-way simulation, but results in a period (years 400-464) in which ablation and accumulation areas are
similar in both simulations.





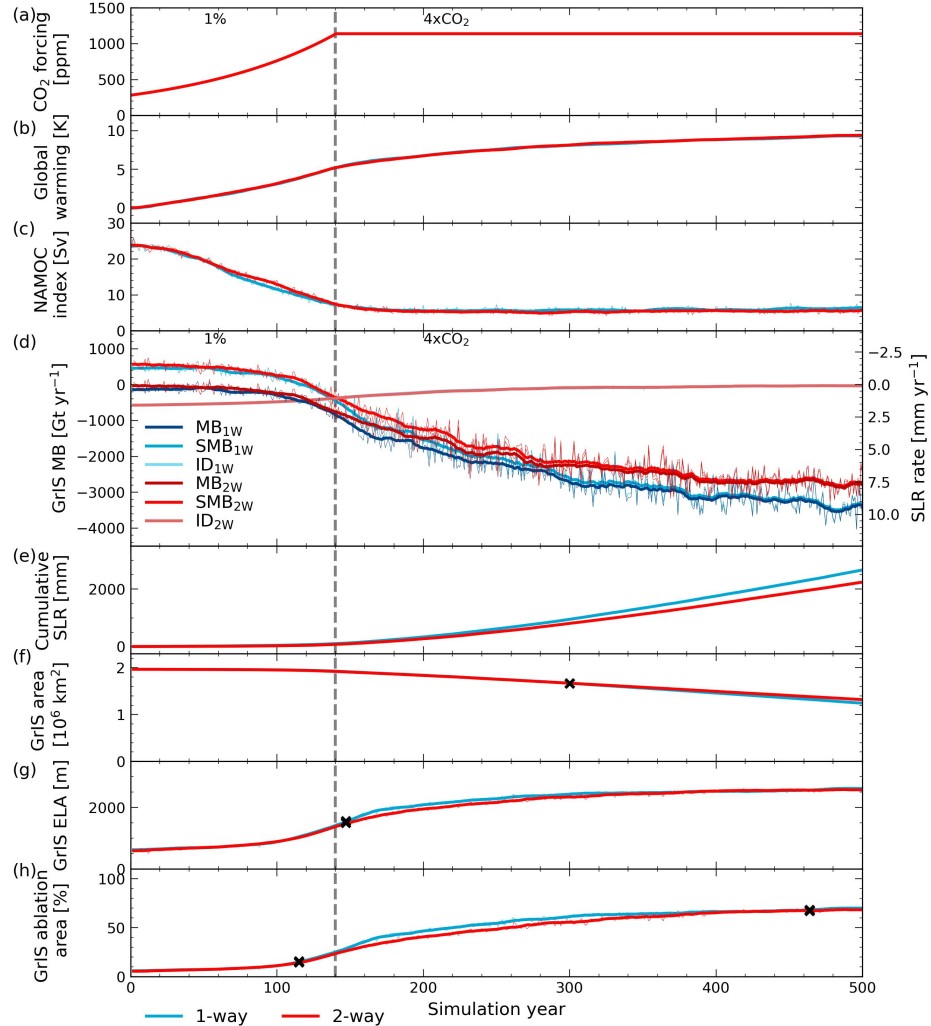

**Figure 2.** Comparison of forcing, climate and ice sheet evolution of 1-way (blue) and 2-way (red) coupled simulations for **(a)** $CO_2$ forcing [ppm], **(b)** global warming [K], **(c)** NAMOC index [Sv], **(d)** GrIS total mass balance (MB) [Gt $yr^{-1}$] and its components ice discharge (ID) and surface mass balance (SMB) (the basal mass balance is not displayed as its contribution is limited), **(e)** GrIS induced cumulative sea level rise [mm], **(f)** GrIS area [$10^6$ $km^2$], **(g)** GrIS equilibrium line altitude [m] and **(h)** GrIS ablation area compared to total GrIS area [%]. The mass balance (components) in **(d)** can be directly converted to the GrIS-induced SLR rate [mm $yr^{-1}$]. The grey dashed line indicates the end of the 1% $CO_2$ ramp-up period and the start of the continuous 4x$CO_2$ period. If the differences between the 1- and 2-way coupled simulations become significant throughout the simulation, the first year of significant difference is marked with a black cross. In **(e)** the year of significance for the annual SLR is given. In **(g, h)** the year of significant difference before the convergence between the two simulations is given as well.



## 4   Climate feedback response to a dynamic GrIS topography

The differences in the mass balance evolutions of the two simulations indicate that the local climate and GrIS surface processes are affected by the coupling. We first consider the positive melt-elevation feedback, which enhances mass loss, and assess its representation in the 1-way coupled configuration, as we compute the climate using a fixed topography in the 1-way simulation. 240   Then, we analyze feedbacks related to atmospheric processes, by considering atmospheric circulation, clouds and precipitation.

### 4.1   The melt-elevation feedback

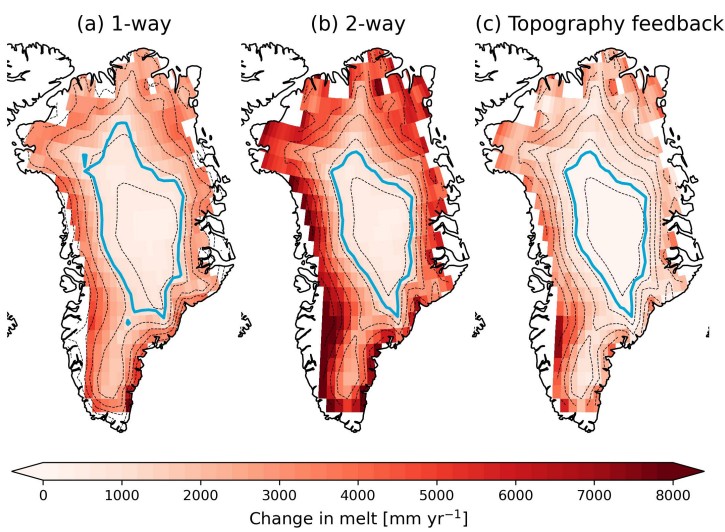

**Figure 3.** Change in melt [mm yr$^{-1}$] as modeled by CLM5 for years 480-500 with respect to the control simulation over the ice sheet extent of the 2-way simulation for the **(a)** 1-way and **(b)** 2-way coupled simulation. **(c)** difference between **(a)** and **(b)**, showing the effect of topographic change on melt evolution. The black dashed lines depict contour lines for every 500 m elevation of **(a)** the 1-way and **(b-c)** the 2-way simulation. The light blue line represents the ELA of **(a)** the 1-way and **(b-c)** the 2-way simulation. All changes are significant according to the emergence criterion (subsection 2.5).

The melt response to the $CO_2$ forcing results in a lowering of the topography. This triggers the melt-elevation feedback (Figure 3), as lower elevations result in higher temperatures, and consequently enhances the melt-albedo feedback, as bare ice and melting snow have a lower albedo. In the 2-way simulation, we account for this, by allowing the topography to evolve in 245   CLM5, while in the 1-way simulation, the topography is constant. The effect of incorporating a dynamic topography on the melt becomes significant in the year 215 and accounting for this results in 66% more melt in CLM5 in the 2-way simulation by year 500. Since both simulations include an interactive calculation of the albedo, the melt differences include both the melt-elevation and the resulting enhancement of the melt-albedo feedback. Next to that, the melt is affected by atmospheric-related feedbacks as well.



The 2 m air temperature ($T_{2m}$) increase resulting from elevation changes causes changes in the GrIS surface energy fluxes. To assess the effect of the elevation change on the $T_{2m}$ and the downwelling longwave radiation ($LW_{in}$), we compute lapse rates (Figure 4 (monthly means), Figure A2 (seasonal maps)) and compare them with the applied lapse rates in the 1-way coupled simulation (Table 1). The applied lapse rate of -6 K km$^{-1}$ corresponds relatively well to the annual mean temperature lapse rate (Table 1). However, applying these pre-defined lapse rates leads to an overestimation of the $T_{2m}$ in summer and an
underestimation in winter. The overestimation in summer is partly compensated by not applying an $LW_{in}$ lapse rate, resulting in an underestimation of $LW_{in}$ in the 1-way coupled simulation. In summer, the surface in the ablation area, which is the area in which most of the elevation change happens, is at the melting point (Figure 4). Therefore, part of the available energy will be used for melting instead of heating the atmosphere, leading to a limited $T_{2m}$ increase in the ablation area. The $LW_{in}$ lapse rates show a similar pattern, as these are largely influenced by atmospheric temperatures.

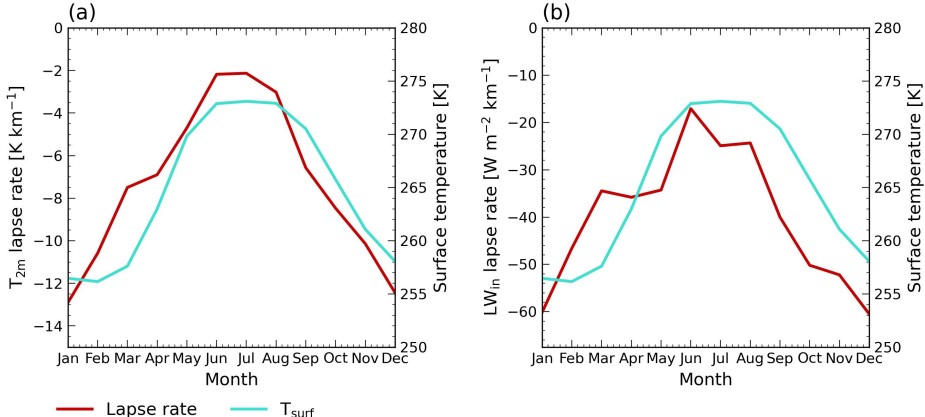

**Figure 4.** Monthly mean lapse rates (dark red) for **(a)** 2m air temperature [K km$^{-1}$] and **(b)** incoming longwave radiation [W m$^{-2}$ km$^{-1}$] for years 480-500. The light red line shows the monthly mean surface temperature [K] of the locations considered for computing the lapse rates and is the same in both **(a)** and **(b)**. The locations considered for the mean lapse rates and surface temperatures are shown in Figure A2.

**Table 1.** 2 m air temperature [K km$^{-1}$] and downwelling longwave radiation [W m$^{-2}$ km$^{-1}$] lapse rates for annual, summer (June, July, August) and winter (December, January, February) means, compared to the applied lapse rates in the 1-way coupled simulation, computed using the years 480-500. The number within brackets represents one standard deviation in the spatial domain.

| Mean lapse rate | $T_{2m}$ | $LW_{in}$ |
|---|---|---|
| Annual | -7.3 (2.4) K km$^{-1}$ | -40.0 (19.4) W m$^{-2}$ km$^{-1}$ |
| Summer (JJA) | -2.4 (1.1) K km$^{-1}$ | -22.1 (13.9) W m$^{-2}$ km$^{-1}$ |
| Winter (DJF) | -12.0 (4.1) K km$^{-1}$ | -55.8 (26.4) W m$^{-2}$ km$^{-1}$ |
| Applied in 1-way | -6 K km$^{-1}$ | 0 W m$^{-2}$ km$^{-1}$ |



To assess the effect of the difference in applied lapse rates, we consider a point that transitions from the accumulation to the ablation area (66.44° N, 45° E, shown in Figure A2). Here, the 1- and 2-way simulations show a similar response as long as the elevation does not change (Figure 5). As soon as the mean summer surface temperature reaches melting point (Figure 5c) and elevation starts to lower (Figure 5a), $T_{2m}$ and melt responses start to diverge between the simulations. Since in the 1-way simulation, the temperature and surface fluxes correspond to a fixed elevation, melt does not increase as strongly as in the 2-way case. However, adding the temperature change due to the lowering of the elevation by considering the -6 K km$^{-1}$ lapse rate (dashed blue in Figure 5c) results in a large overestimation of the 1-way $T_{2m}$, potentially causing an overestimation of the sensible and latent heat flux contribution to the melt energy. Although this is partially compensated by an underestimation of the incoming longwave flux, this could result in an overestimation of the melt-elevation and melt-albedo feedback when computing mass loss in CISM2, which could explain part of the overestimated mass loss.

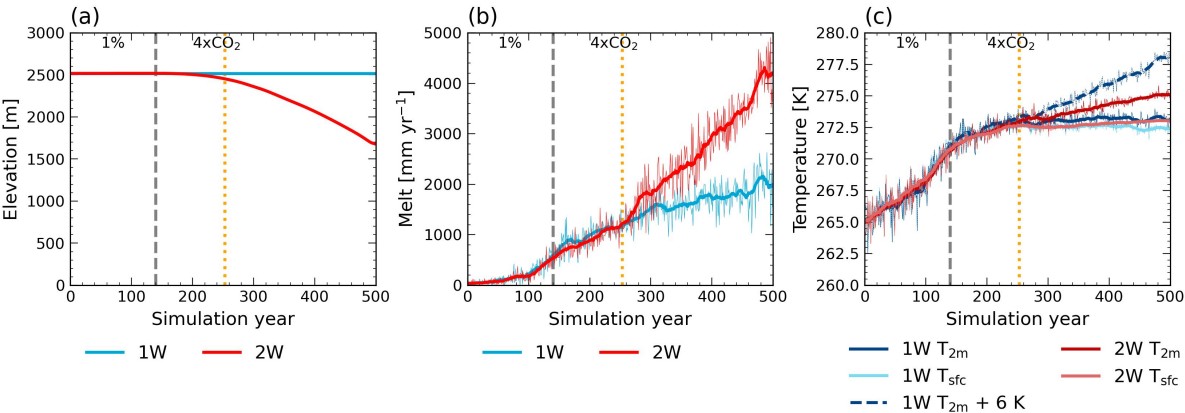

**Figure 5.** Evolution of **(a)** elevation [m], **(b)** annual melt [mm yr$^{-1}$] and **(c)** mean summer (June, July, August) 2 m air and surface temperature [K] for the 1-way (blue, 1W) and 2-way (red, 2W) coupled simulations for a point that transitions to the ablation area (66.44° N, 45° E, shown in Figure A2). The grey dashed line indicates the end of the 1% $CO_2$ ramp-up period and the start of the continuous 4x$CO_2$ period. The orange dashed line indicates the timing of reaching surface melt conditions throughout the whole summer (273 K). In **(c)** the blue dashed line shows the evolution of 1-way $T_{2m}$ when applying the -6 K km$^{-1}$ lapse rate to the 2-way elevation change.

## 4.2 Elevation feedbacks related to changes in precipitation, atmospheric circulation and clouds

Besides affecting temperature, changes in the GrIS topography can change the local climate in several other ways. First of all, the effect of the presence of the cold and high-elevation GrIS on atmospheric circulation might change as its surface lowers and warms, which might have large implications for the regional climate. Atmospheric blocking events, associated with persistent high-pressure conditions over Greenland, can be related to higher summer temperatures and reduced cloud cover over Greenland (Hofer et al., 2017). Therefore, increases in the amount of summer atmospheric blocking have been linked to increased melt (Hanna et al., 2022). We find a large reduction of Greenland blocking resulting from a changing GrIS surface




(Figure 6a, b), especially in summer. This aligns with the positive relationship between orography and blocking events (Mullen, 1989; Narinesingh et al., 2020; Sellevold et al., 2022). The changing GrIS geometry has no significant effect on the phase of the NAO (Figure 6d, e). However, the summer NAO index shows a trend towards a positive phase as a result of the warming in both simulations (p < 0.001).

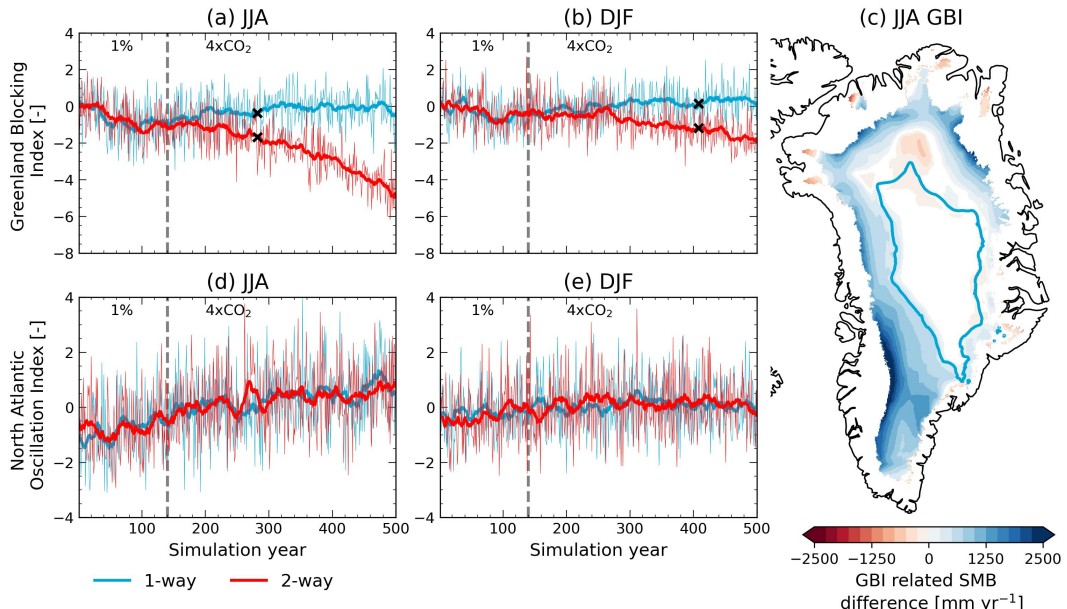

**Figure 6.** Evolution of **(a, b)** Greenland blocking index (GBI) [-] and **(d, e)** North Atlantic oscillation (NAO) index [-] for **(a, d)** June, July, August (JJA) mean and **(b, e)** December, January, February (DJF) mean for the 1-way (blue) and 2-way (red) coupled simulations. The grey dashed line indicates the end of the 1% $CO_2$ ramp-up period and the start of the continuous 4x$CO_2$ period. If the differences between the 1- and 2-way coupled simulations become significant throughout the simulation, the first year of significant difference is marked with a black cross. **(c)** shows the difference in SMB [mm yr$^{-1}$] caused by the coupling that can be explained by the difference in summer blocking, computed using linear regression for the area with significant Pearson correlation ($p = 0.01$) for the years 480-500, for grid points that have not deglaciated in both simulations. The light blue line represents the ELA in the 2-way simulation in years 480-500.

To relate the decreased blocking to the projected mass loss, we regress the GBI differences between the 1- and 2-way coupled simulation onto the SMB differences (from CISM2, Figure 6c). A positive regression coefficient indicates a linear relationship between the differences in blocking and the differences in SMB reduction and its magnitude represents the slope of this linear relationship. The relationship between the GBI and SMB differences is strongest in the ablation area. In the accumulation area, the surface melt is limited, which likely is the reason that we do not find a strong relationship with changes in Greenland blocking. As the topographies of the two simulations evolve differently, so does the SMB pattern, resulting in small areas in which the correlation signal is negative, indicating a slightly larger SMB reduction in the 2-way coupled simulation. This might not be caused by the differences in blocking, but rather by differences in the topography. Considering areas with significantly




strong correlation (positive or negative, as tested with a t-test using $p = 0.01$), we can link 49% of the difference in SMB

reduction between the two simulations at the final simulation years with the decrease in atmospheric blocking (Figure 6c). As the SMB differences are mainly dominated by melt differences in this period, we hypothesize that the decrease in blocking as a result of GrIS surface elevation changes (Figure A1a-e) acts as a negative feedback on surface melting.

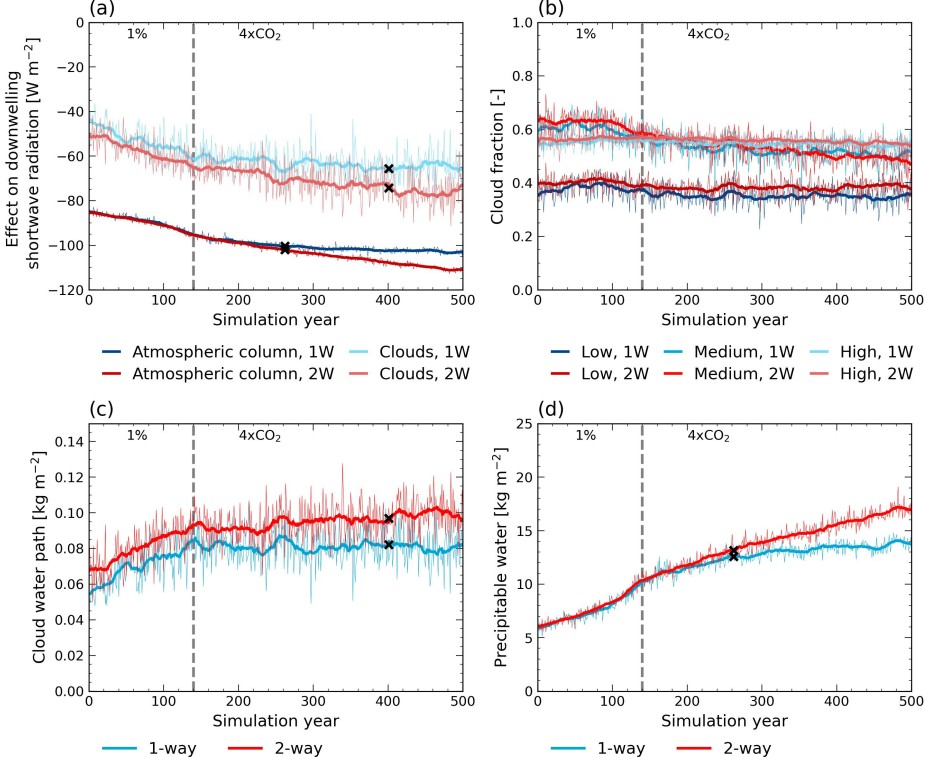

**Figure 7.** Timeseries over the first year ice sheet extent of 1-way (blue, 1W) and 2-way (red, 2W) simulations for **(a)** effect on downwelling shortwave radiation [W m$^{-2}$] of the atmospheric column and clouds. The effect of the atmospheric column is computed by comparing the incoming solar radiation at the top of the atmosphere with the received radiation at the surface for cloud-free conditions. The effect of clouds is computed by comparing the incoming solar radiation at the top of the atmosphere with the received radiation at the surface for all-sky conditions and correcting for the effect of the atmospheric column. **(b)** cloud fraction [-] for low-, medium- and high-level clouds, **(c)** cloud water path [kg m$^{-2}$] and **(d)** precipitable water in the atmospheric column [kg m$^{-2}$]. The grey dashed line indicates the end of the 1% CO$_2$ ramp-up period and the start of the continuous 4xCO$_2$ period. If the differences between the 1- and 2-way coupled simulations become significant throughout the simulation, the first year of significant difference is marked with a black cross.

We find another negative feedback, this time related to clouds and water vapor. Under increasing temperatures, the atmosphere can contain more moisture (as described by the Clausius-Clapeyron relationship), which can lead to an increase in

clouds and enhanced precipitation (Pall et al., 2007). The amount of precipitable water in the atmospheric column (Figure 7d) increases in both simulations, although stronger in the 2-way coupled simulation, as the atmospheric column warms more




and becomes larger as a result of elevation change. The increase in precipitable water translates to a stronger reflection and absorption of incoming solar radiation in the atmosphere (Figure 7a). The differences in the atmospheric effect on incoming shortwave radiation and precipitable water both become significant in year 262, indicating they are strongly related. There is no increase in cloud cover (Figure 7b) in both simulations and even a slight decrease in medium-level clouds. However, cloud cover itself does not strongly control the incoming shortwave radiation. Instead, the cloud optical thickness, influenced by the presence of liquid water and ice in the atmosphere, is an important determining factor for the incoming shortwave radiation (Ettema et al., 2010). As the cloud water path (Figure 7c) increases throughout the simulation, meaning clouds become thicker, more shortwave radiation is reflected. This effect is stronger than the effect resulting from the slight decrease in cloud cover, resulting in a stronger cloud effect on incoming surface solar radiation (Figure 7a) in the 2-way coupled simulation. The differences in cloud water path and cloud effect on downwelling shortwave radiation both become significant in year 401, showing that the cloud thickness is the determining factor for incoming shortwave radiation, rather than cloud cover.

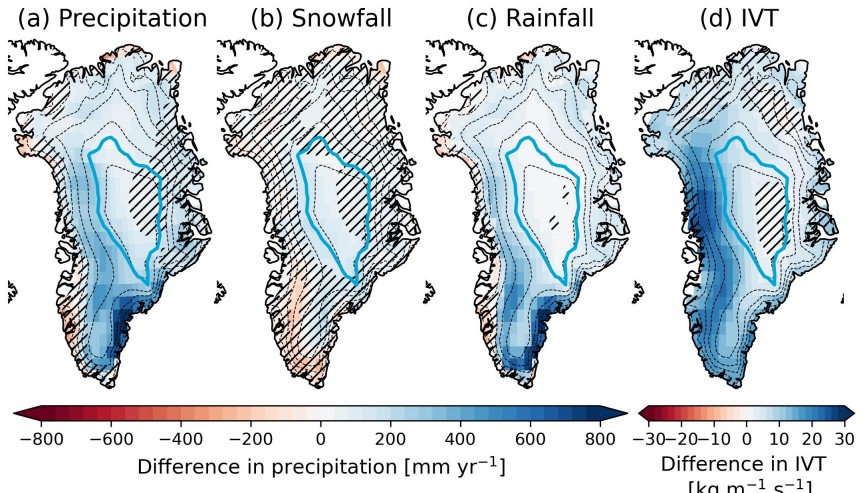

**Figure 8.** Annual mean differences (2-way minus 1-way) resulting from the coupling for **(a)** total precipitation [mm yr$^{-1}$] (the sum of snowfall and rainfall) **(b)** snowfall [mm yr$^{-1}$], **(c)** rainfall [mm yr$^{-1}$] and **(d)** mean integrated vapor transport (IVT) [kg m$^{-1}$ s$^{-1}$] for years 480-500 over the first year ice sheet extent. The dashed black lines depict contour lines for every 500 m elevation of the 2-way simulation. The light blue line represents the ELA of the 2-way simulation. Hatches denote areas in which the differences between the 1- and 2-way coupled simulations have not become significant before the year 500.

Finally, a lowering topography results in an overall increase in precipitation, especially in the southeast (Figure 8a). Higher temperatures in the 2-way simulation result in more evaporation from the surrounding oceans. An increase in the transport of this moisture leads to increased precipitation over the GrIS (Figure 8d). In some locations in the western and northern margins, precipitation decreases, although not significant in all locations, as orographic precipitation moves more land inward, following the margin. In the western margins, we see a corresponding strong increase in IVT. The difference in precipitation is mainly caused by increased rainfall (Figure 8c), rather than increased snowfall (Figure 8b). Snowfall moves more towards the interior,





as temperatures in the margins become too high for precipitation to fall as snow. As a result, snowfall decreases in the ablation
area and increases in the accumulation area (Figure A3). The increase in snowfall in the interior contributes positively to the
SMB, acting as a negative feedback. However, the elevation effect on temperature results in a larger fraction of rainfall, which
can enhance the melt-albedo feedback. Therefore, the changes in rainfall can be considered a positive feedback.

## 5  GrIS mass balance and climate response to the reintroduction of PI $CO_2$ conditions

The 4x$CO_2$ forcing triggers large responses in global, Arctic and GrIS temperatures (Figure 2b, Figure 9b). By year 350, the
GrIS has a strongly negative mass balance in the 2-way coupled simulation (Figure 2d, Figure 9d) causing an SLR rate of 6.6 $\pm$
1.0 mm yr$^{-1}$, with a cumulative GrIS SLR contribution of 1.13 m (Figure 2e, Figure 9e). The discharge contribution to the mass
balance is limited (-78 $\pm$ 8.8 Gt yr$^{-1}$), as the ice sheet has strongly retreated. Besides, the NAMOC has collapsed (Figure 2c,
Figure 9c), reducing the amount of northward heat transport. We branch another simulation from the 2-way coupled simulation,
starting in year 350, and apply an annual 5% $CO_2$ reduction until PI $CO_2$ conditions are reached in year 377 (Figure 9a) to
assess the GrIS mass balance and climate response to $CO_2$ reduction. We first consider global, Arctic and North Atlantic climate
change and evaluate GrIS mass loss response to $CO_2$ reduction. We zoom in to the first century after the $CO_2$ reduction, which
is characterized by a complex transition phase towards a colder climate, and look at the NAMOC evolution and its impact on
regional climate. Finally, we assess GrIS surface processes by considering the response of the snowpack.

The response to $CO_2$ reduction can be divided into two periods. During the first period, spanning the 27 years during the
$CO_2$ ramp-down and the following 85 years, the GrIS experiences a complex transitional phase because of strong interactions
with the NAMOC. During this phase, the NAMOC is still weak, as its recovery is delayed compared to the atmospheric
response, resulting in relatively cold temperatures in the Arctic and over the GrIS (Figure 9b). Under these largely reduced air
temperatures resulting from the $CO_2$ decrease and weak state of the NAMOC, the amount of GrIS melt decreases strongly,
resulting in a small positive SMB and MB (Figure 9d). During the $CO_2$ ramp-down and first years thereafter, there is an
additional SLR of 118 cm. This is followed by a short period (year 413 to 461) of positive mass balance, leading to a 13 cm
sea level drop, after which GrIS SLR contribution is halted (Figure 9d, e). The second period, spanning from the end of the
transitional phase (year 462) to the end of the simulation, is characterized by initial temperature increases in the Arctic and over
the GrIS as a response to an overshooting rebound of the NAMOC, followed by a slow continuous decrease in atmospheric
temperature and NAMOC strength. The mass balance becomes smaller than during the transitional phase, nearing zero, and has
recovered to its PI state (0.04 $\pm$ 0.23 mm yr$^{-1}$) by the year 452 (Table B1). However, from year 735 onward, the mass balance
becomes slightly positive again, leading to an annual sea level drop of 0.06 $\pm$ 0.19 mm yr$^{-1}$ at the end of the simulation. This
indicates that there might be potential for ice sheet regrowth, although at this rate this would take over ten thousand years.
At the end of the simulation, the SMB has not recovered (Table B1), as global and GrIS temperatures remain elevated at the
end of the simulation compared to the initial state. However, the retreated ice sheet margins result in a smaller contribution of
ice discharge to the mass balance (Figure 9d). Together with a small positive SMB, the small negative contribution of the ice
discharge allows for a small positive total mass balance. We compare this state to the state of the GrIS under the same global





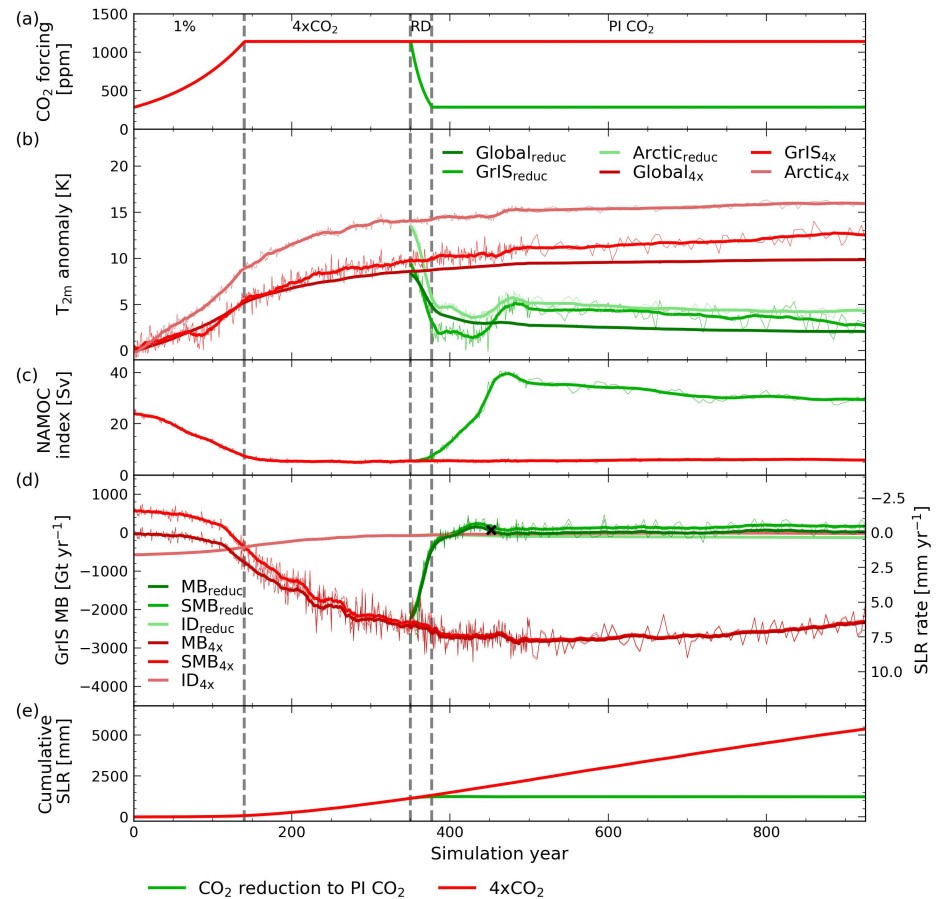

**Figure 9.** Evolution of $CO_2$ reduction to pre-industrial $CO_2$ (green, reduc) and the full $4xCO_2$ (red, 4x) simulations for **(a)** $CO_2$ forcing [ppm], **(b)** global, evolving GrIS and Arctic near-surface air temperature ($T_{2m}$) anomalies with respect to PI [K], **(c)** North Atlantic Meridional Overturning Circulation (NAMOC) index [Sv], **(d)** GrIS total mass balance (MB) [Gt yr$^{-1}$] and its components ice discharge (ID) and surface mass balance (SMB) (the basal mass balance is not displayed as its contribution is limited) and **(e)** GrIS cumulative sea level rise contribution [mm]. The mass balance (components) in **(d)** can be directly converted to the GrIS-induced SLR rate [mm yr$^{-1}$]. The grey dashed lines indicate the end of the 1% $CO_2$ ramp-up, the continuous $4xCO_2$, the ramp-down (RD) to PI $CO_2$ and the continuous PI $CO_2$ periods. If a variable has recovered throughout the simulation, the first year of recovery is marked with a black cross.

$T_{2m}$ anomaly (2 K) during the $CO_2$ ramp-up (year 70), where we see an ice sheet that is largely marine-terminating and is out of balance as a result of this temperature anomaly (Table 2). Although the ice sheet has a stronger positive SMB in year 70, the large ice discharge leads to net mass loss, while for the retreated ice sheet at the end of the $CO_2$ reduction simulation, the total
mass balance is slightly positive, despite the smaller SMB.

During the $4xCO_2$ forcing period, the North Atlantic warms less rapidly due to the weakening of the NAMOC, resulting in a warming hole southeast of Greenland (Figure 10a). During the transitional $CO_2$ reduction phase, the North Atlantic cools



**Table 2.** Comparison of the state of the GrIS under an annual global $T_{2m}$ anomaly of 2.0 K during the $CO_2$ ramp-up period in year 70 (average over years 60-80) and at the end of the $CO_2$ reduction simulation (average over years 825-925). The number within brackets represents one standard deviation.

|  | Year 70 (60-80 mean) | End of simulation (825-925 mean) |
| --- | --- | --- |
| Global $T_{2m}$ anomaly | 2.0 (0.22) K | 2.0 (0.08) K |
| GrIS $T_{2m}$ anomaly | 1.9 (0.68) K | 2.9 (1.0) K |
| Total mass balance | -105 (76) Gt $yr^{-1}$ | 21 (70) Gt $yr^{-1}$ |
| SMB | 438 (81) Gt $yr^{-1}$ | 161 (70) Gt $yr^{-1}$ |
| Integrated SMB | 224 (40) mm $yr^{-1}$ | 103 (43) mm $yr^{-1}$ |
| Ice discharge | -520 (7.0) Gt $yr^{-1}$ | -124 (4.8) Gt $yr^{-1}$ |

rapidly as this warming hole persists under a weak NAMOC (Figure 10c). However, the delayed NAMOC overshoot causes the North Atlantic temperatures to rise subsequently, causing the warming hole to disappear (Figure 10e). Besides, the timing of

changes in GrIS, North Atlantic and Arctic processes strongly coincides with the timing of NAMOC index changes (Figure B2), highlighting the complexity of the interactions between the ocean, atmosphere, Arctic sea ice and the GrIS.

The Arctic interacts strongly with the NAMOC and the atmosphere. Due to a large drop in Arctic $T_{2m}$ as a response to reverting $CO_2$ conditions, the March Arctic sea ice extent recovers quickly (Figure 10b), amplified by the thin ice feedback (Notz, 2009), as an area with thin sea ice expands faster. In contrast, the September sea ice does not recover, since the newly

formed sea ice is relatively thin (Figure 10d) and ocean and atmosphere summer temperatures remain elevated. Besides, the sea ice melt is amplified by the sea ice-albedo feedback (Curry et al., 1995), leading to the nearly complete melting of the recently formed sea ice in summer. The regrowth of sea ice in winter influences the NAMOC strength, as brine rejection leads to more saline high-latitude waters, enhancing overturning. The overshooting response of the NAMOC leads to a subsequent decrease in Arctic sea ice extent and thickness as a result of increased northward heat transport.

The mixed layer depth (MLD, Figure 10f), which is an indicator for the amount of overturning and thus for the strength of the NAMOC, decreases in the Labrador Sea, Irminger Sea and the Norwegian Sea during the 1% $CO_2$ ramp-up period. In contrast, the MLD in the Nansen Basin starts to increase slightly. This indicates a movement of the area with strong overturning towards higher latitudes, possibly caused by the decrease in Arctic sea ice extent (Figure 10b). The MLDs in the Labrador Sea, Irminger Sea and Norwegian Sea all experience an overshoot after the $CO_2$ ramp-down, similar to the NAMOC index. Only

in the Labrador Sea, where we see the large remaining temperature anomalies as well (Figure 10c) - indicating elevated heat transport, there is no complete recovery from the overshoot.

Looking at GrIS surface processes, we find that melt increases due to the $CO_2$ forcing, are partly countered by enhanced refreezing (Figure 11a). At the end of the ramp-up period, in the year 130, the peak in the amount of refreezing is reached, despite the ongoing increase in the amount of available water for refreezing. This implies that around this year the pore space

and/or available energy for melt has started to decrease. This peak refreezing behavior is typical for high emission scenarios





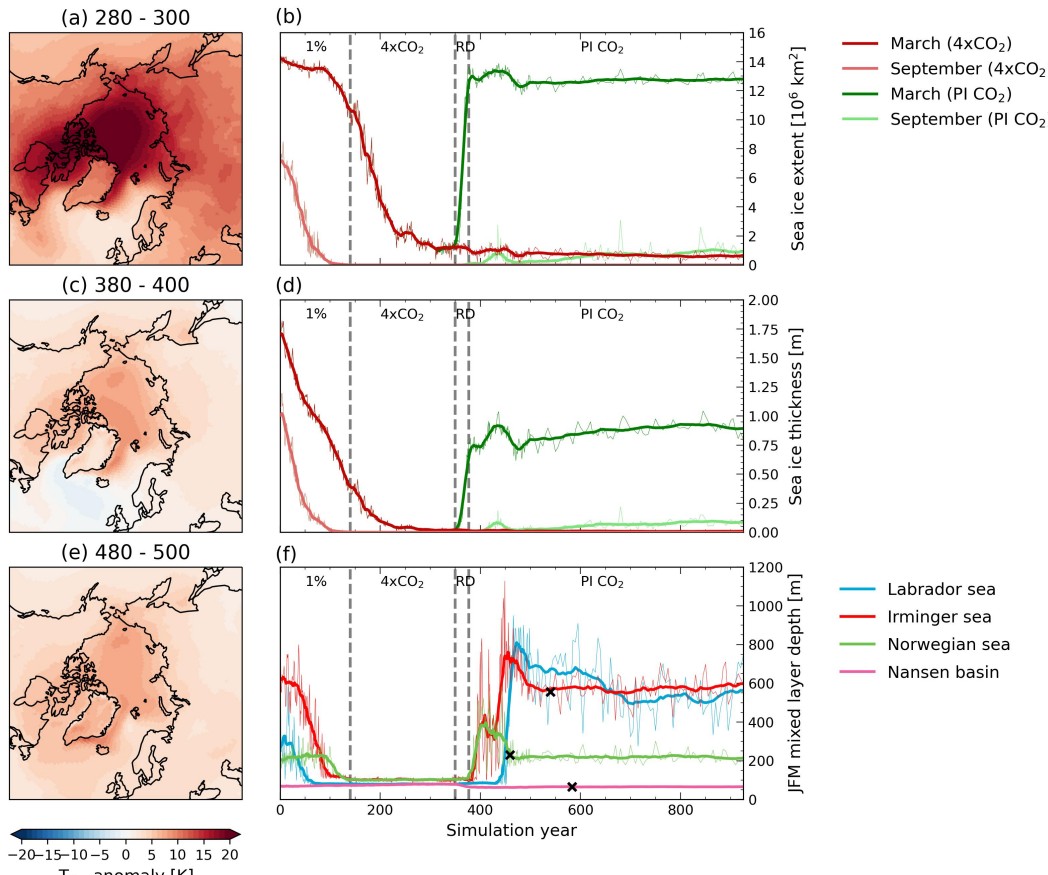

**Figure 10. (a, c, e)** Mean anomalies of near-surface temperature ($T_{2m}$) [K] for the $CO_2$ reduction simulation compared to the control simulation for **(a)** years 280-300 during the constant $4xCO_2$ period, **(c)** years 380-400 just after the $CO_2$ ramp-down in the transition period and **(e)** years 480-500 after the transition period. **(b, d)** Evolution of reduction to pre-industrial $CO_2$ (green) and the full $4xCO_2$ (red) simulations for **(b)** sea ice extent [$10^6$ km$^2$] and **(d)** sea ice thickness [m] of the evolving sea ice extent in March and September. **(f)** Evolution of the mean mixed layer depth (MLD) of January, February and March (JFM) in the Labrador Sea (blue), Irminger Sea (red), Norwegian Sea (green) and Nansen Basin (pink) for the reduction to pre-industrial $CO_2$ simulation. In **(b, d, f)** the grey dashed lines indicate the end of the 1% $CO_2$ ramp-up, the continuous $4xCO_2$, the ramp-down (RD) to PI $CO_2$ and the continuous PI $CO_2$ periods. If a variable has recovered throughout the simulation, the first year of recovery is marked with a black cross.

(Noël et al., 2022). The refreezing capacity peaks earlier, as the snowpack is not able to refreeze a similar fraction of the larger amount of available water, since it does not become thicker concurrently (Figure B3a). After the $CO_2$ ramp-down, the amount of refreezing slightly peaks again in year 478, as the snowpack partially recovers. However, the poorer state of the snow compared to its state before the $4xCO_2$ forcing was applied, characterized by higher snow temperatures and a thinner 380 snowpack (Figure B3), prevents the snowpack from returning to refreezing rates similar to year 130, despite the similar amount





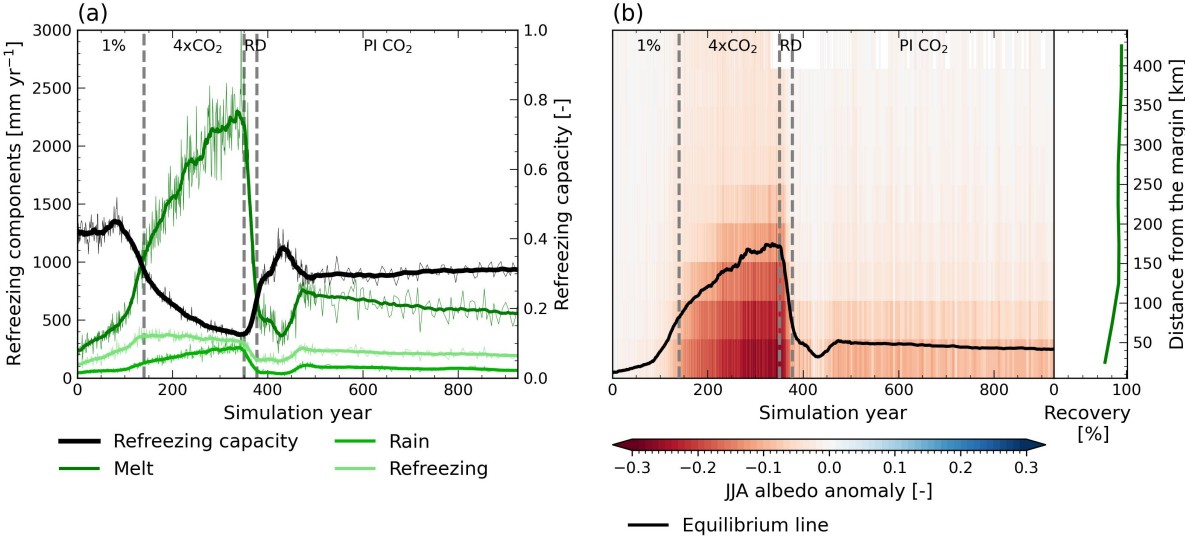

**Figure 11.** Evolution of the snow properties **(a)** refreezing capacity (black) [-], defined as the refreezing divided by the amount of available water for refreezing (meltwater and rain) and its components meltwater (dark green), rainfall (green) and refreezing (light green) [mm yr$^{-1}$] over the evolving ice sheet and **(b)** summer (JJA) albedo anomalies [-] with respect to the control simulation as a function of time and distance to the ice sheet margin for the reintroduction of PI $CO_2$ simulation and the location of the equilibrium line (black). In **(b)**, the dark green line shows the relative recovery (in %) for each distance class. The grey dashed lines indicate the end of the 1% $CO_2$ ramp-up, the continuous 4x$CO_2$, the ramp-down (RD) to PI $CO_2$ and the continuous PI $CO_2$ periods. None of the variables in **(a)** have recovered by the end of the simulation.

of available water. Besides, the poorer state of the snowpack, combined with a larger amount of available water than in the initial state, results in a refreezing capacity that does not recover (Table B1).

Next to the refreezing, the ice sheet albedo does not completely recover either (Figure 11b). In the cooler period around years 400 to 450, the albedo nearly recovers to its initial state, even around the margins, as melt strongly reduces. However, 385 subsequent temperature increases due to the NAMOC recovery lead to more melt and rain and a corresponding decrease in albedo, aligning with the expansion of the ablation area, resulting in an ELA that does not recover. The lower surface albedo causes the net shortwave flux to remain larger, which is the largest contributor to the fact that the SMB and SEB (Figure B1) do not recover completely.

## 6 Discussion

The results presented in this study highlight the importance of accounting for interactions between the GrIS and the climate for SLR projections. We find that uncoupled simulations of ice sheet and climate evolution can lead to an overestimation of the SLR contribution. For CESM2, we find an overestimation of the GrIS SLR contribution of 17% under a 4xPI $CO_2$



forcing when using the 1-way coupled configuration. Similarly, Ridley et al. (2005) found an overestimation of mass loss in an uncoupled AOGCM-ISM simulation, although these only become apparent after a total mass loss of around 2.5 m SLE, opposed to 0.7 m SLE in our study. Our results are close to those of Gregory et al. (2020), who found a relative overestimation of mass loss of 13% in an uncoupled abrupt 4xCO$_2$ simulation using the Glimmer ice sheet model coupled to FAMOUS-ice. Gregory et al. (2020) attributed part of the smaller mass loss in the coupled simulation to a larger precipitation increase, as precipitation in the southwest moves land inward, which is similar to what we find. However, the larger amount of rainfall does not contribute positively to the SMB and although the snowfall in the accumulation area is larger in the 2-way coupled simulation, its contribution is limited (28 Gt yr$^{-1}$ by year 500, Figure A3). Besides, Gregory et al. (2020) related the smaller mass to a negative cloud feedback on the downwelling shortwave radiation due to a larger cloud fraction in the coupled simulation. Although we do not find significant differences in cloud fraction, we find significantly thicker clouds in the 2-way coupled simulation. Ettema et al. (2010) indicated that cloud thickness, rather than cloud cover is the most important cloud-related control on incoming shortwave radiation, as cloud cover is not a measure of transmissivity, which aligns with our results but is contrasting to Gregory et al. (2020).

This study is the first to look into the effect of GrIS topographic changes on atmospheric blocking. In our 2-way coupled simulation, blocking is significantly reduced when the GrIS topography lowers. Besides, we find a strong relationship between atmospheric blocking and SMB evolution, linking nearly half of the difference in SMB reduction to the decrease in blocking. Therefore we hypothesize that the topographic control on Greenland blocking occurrence can act as a negative feedback on surface melt. However, further investigation into the causes of these changes in blocking and their relationship with melt is necessary to make a definite attribution. Observations and climate projections show that blocking has a strong effect on GrIS melting (Sellevold and Vizcaíno, 2020; Hanna et al., 2022). However, Hanna et al. (2018) showed that climate models are not able to capture the present-day increase in GBI and consistently project a future decrease in blocking. Therefore blocking projections should be treated with caution. Nevertheless, we relate the significant decrease in GBI in our 2-way simulation to large topographic changes, which are not the main driver of present-day GBI changes.

Besides, we hypothesize that a large part of the increased mass loss in the 1-way simulation arises from the application of a uniform T$_{2m}$ lapse rate. Compared to the lapse rates computed from our 1- and 2-way simulations, the applied uniform lapse rate results in an overestimation of summer T$_{2m}$ in the ablation area and therefore melt might be overestimated. Although the use of uniform lapse rates in offline simulations is common (e.g. Aschwanden et al., 2019; Sellevold and Vizcaíno, 2020; Bochow et al., 2023), it has been pointed out that lapse rates are not temporally and spatially uniform (Hanna et al., 2005; Gardner et al., 2009) and that the melt-elevation feedback is sensitive to the choice of the lapse rate (Zeitz et al., 2022). Crow et al. (2024) found that applying a seasonally and spatially varying lapse rate for downscaling SMB from 1-way coupled CESM simulations of the MIS-11c Greenland Ice Sheet gives the most accurate representation of the melt-elevation feedback, showing that it is likely that offline corrections in ice sheet models can be improved by accounting for the temporal and spatial variability of the lapse rate in the ablation area, by considering the surface temperature. It should be noted that our computed T$_{2m}$ lapse rates are not only affected by the surface reaching melting point but also by the other negative feedbacks influencing temperature, like increasing cloud thickness and reduced atmospheric blocking.



When we apply a 5% $CO_2$ reduction to PI $CO_2$, we find that the GrIS SLR contribution can be halted. However, the chosen $CO_2$ forcing, the timing and the rate of the $CO_2$ ramp-up and ramp-down likely have a large influence on the mass balance
evolution and influence whether mass loss can be halted or reversed. In our simulation, a small positive mass balance, during the transitional phase just after $CO_2$ reduction, as well as at the end of the simulation, allows for a small regrowth of the ice sheet. However, reversing the 1.1 m (equal to 14% of the initial ice sheet volume) of mass lost during the forcing period would likely take ten-thousands of years. Our simulation is designed to investigate the response of the GrIS to $CO_2$ reduction and the climate interactions that play a role therein, rather than finding thresholds for irreversible mass loss and potential equilibrium
states, in contrast to previous work. Despite the long time scales used in previous work, the importance of certain feedbacks and interactions has been pointed out before. Using a coupled AOGCM-ISM configuration, Ridley et al. (2010) investigated the potential regrowth of the GrIS from 11 different initial states when reverting to PI climate. They showed that for states with an ice sheet area reduction of 10% and 20%, which is in the same order as this study, climate-ice sheet interactions, resulting from the temperature dependence on elevation, make the difference in whether or not the ice sheet could regrow. Although in the
study by Ridley et al. (2010) the ice sheet is coupled to the ocean model, interactions with the NAMOC are likely not captured due to the experimental set-up with asynchronous coupling. Using the FAMOUS-ice - GLIMMER coupled configuration, Gregory et al. (2020) performed $CO_2$ removal experiments starting from multiple GrIS (steady) states and highlighted the importance of the melt-albedo feedback on reversibility, which our results agree with. Their mass loss trajectories do not show a complex transitional phase as the model set-up does not allow for interaction with the ocean, meaning there is no
influence from an overshooting NAMOC recovery. Bochow et al. (2023) used two ice sheet models to explore the GrIS mass balance response to a linear temperature ramp-up and subsequent ramp-down. Their experiment with a 100 year ramp-up to 7 K warming followed by a 100 year ramp-down to a remaining 2 K warming and an additional simulated 100 kyr at this 2 K warming level, is closest to our experiment. Depending on the model, they find a notable larger mass loss of 2 to 6 m SLE, although over a larger time scale. However, at the end of our simulation, the ice sheet has a small, but positive mass balance,
and therefore we do not expect the ice sheet to lose more mass if we would extend the simulation. The study by Bochow et al. (2023) only includes limited interactions between the ice sheet and the atmosphere (melt-elevation and melt-albedo feedback and precipitation changes) and no interactions with the ocean. These differences show the large impact of complex climate-ice sheet interactions on projected mass loss.

The large impact of interactions with the NAMOC in the transitional phase towards a colder climate is apparent in this study
and is not discussed in other studies on GrIS mass loss behavior in $CO_2$ reduction scenarios. However, the response of the (N)AMOC to $CO_2$ reduction has been studied under similar 1% ramp-up and ramp-down 4x$CO_2$ scenarios (Wu et al., 2011; An et al., 2021; Oh et al., 2022). There is agreement on the existence of overshoot behavior of the AMOC, reaching AMOC strengths above those in the PI climate, and this response is delayed compared to the atmospheric response to $CO_2$ reduction. This behavior is mainly attributed to the enhanced salt-advection feedback. Our study only focuses on interactions with the
Arctic and North Atlantic. Hence, changes in salinity in the subtropics resulting from an intensification of the hydrological cycle (Wu et al., 2011) are not taken into consideration in our analysis. Therefore, our assessment of NAMOC changes related



to changes in the Arctic and North Atlantic describes interactions between and dependencies on the different processes in these areas, rather than attributing causes and effects of NAMOC changes.

## 7 Conclusions

In this study, we compared the 1- and 2-way coupled configuration of CESM2-CISM2, to investigate GrIS surface-related feedbacks and assess to what extent 1-way coupled simulations can capture the elevation feedbacks. We find that in a 4xCO$_2$ scenario, the sum of topography-related feedback responses results in 66% more melt by year 500 when considering a dynamic GrIS by using 2-way coupling. However, the offline topography correction in CISM2 in the 1-way configuration overestimates these feedbacks, resulting in an additional mass loss of 17% in the 1-way simulation. This overestimation partly arises from the
overestimation of the positive melt-elevation feedback, as well as from not including several negative atmospheric feedback mechanisms. In the 1-way coupled simulation, a uniform temperature lapse rate of -6 K is used to correct for elevation changes. However, the temperature lapse has a large seasonal variability, as the surface in the ablation area is at melting point in summer. Overestimated temperature lapse rates in summer result in an overestimation of the melt-elevation feedback. Taking this seasonal dependency into account when doing offline corrections for elevation could help resolve part of the discrepancy between
the 1- and 2-way coupled simulations. Especially for other models that heavily rely on temperature-related parameterizations, it is of great importance to carefully consider the applied lapse rate, as a uniform lapse rate might result in a large overestimation of melt. Besides enhancing melt, the changes in GrIS topography result in increased precipitation, reduced atmospheric blocking and a decrease in solar radiation reaching the surface resulting from increased atmospheric water vapor content and cloud thickness. These atmospheric responses act as negative feedbacks on melt, resulting in smaller mass loss in the 2-way
coupled simulation. These results stress the importance of considering ice sheet-climate interactions for SLR projections, by using 2-way coupled simulations.

A rapid reduction of atmospheric CO$_2$ from 4xPI to 1xPI conditions leads to strong responses of the global, GrIS and Arctic climate. Despite a remaining global temperature anomaly of 2 K, compared to the initial PI climate, GrIS mass loss halts, resulting from the small contribution of discharge as the ice sheet has partially retreated under a total mass loss of 1.1 m SLE.
The collapsed NAMOC results in a warming hole south of Greenland during the forcing period and the first years after the CO$_2$ ramp-down, but eventually recovers and overshoots under the reintroduced PI CO$_2$, resulting in a cooling hole in this area. This results in a complex transitional phase in which temperatures first decrease rapidly but increase subsequently. Arctic sea ice extent in winter recovers well, but the ice remains thin and largely melts away in summer. As a result, Arctic temperatures remain high. Although the GrIS mass balance recovers, its SMB does not, resulting from elevated temperatures, elevated ocean
heat transport, and a snowpack that does not recover compared to the initial PI state. The refreezing capacity of the snowpack reduces as a result of a smaller snow depth and higher snow temperatures. Besides, the albedo of the snowpack remains lower, leading to a stronger melt-albedo feedback.

This idealized CO$_2$ ramp-down scenario shows the importance of ice sheet-climate interactions on the response of the GrIS mass balance and climate to CO$_2$ reduction. Overshoot scenarios are becoming more important in policy-making, while studies



regarding GrIS mass loss reversibility focus on multi-millennial behavior and do not always account for interactions with the climate (e.g. in: Ridley et al., 2005; Gregory et al., 2020; Bochow et al., 2023). Although this study is a first step towards understanding GrIS mass balance evolution under $CO_2$ reduction, considering shorter-term simulations and less rapid ramp-down scenarios is crucial for SLR projections. In short-term simulations, SLR might not be halted when decreasing $CO_2$ concentrations, as discharge contributes more to the mass balance, and the interactions with the NAMOC could be smaller,

as the magnitude of the NAMOC response to $CO_2$ reduction depends on how large the $CO_2$ forcing is and for how long it is applied (Wu et al., 2011).



## Appendix A: 1- and 2-way coupled simulations

**Figure A1.** Ice thickness [m] and surface mass balance (SMB) [mm yr$^{-1}$] as downscaled to the CISM2 grid (4 km) for **(a, f)** the mean of the control simulation, **(b, c, g, h)** anomalies of the 2-way coupled simulation with respect to the control simulation and **(d, e, i, j)** anomalies of the 2-way with respect to the 1-way coupled simulation for grid points that have not deglaciated in both simulations, for **(b, d, g, i)** the years 240-260 and **(c, e, h, j)** the years 480-500. The dashed black lines depict contour lines for every 500 m elevation of the control **(a, f)** and 2-way coupled **(b-e, g-j)** simulations. Hatches denote areas in which the differences between the 1- and 2-way coupled simulations have not become significant before the last year of the corresponding period. Because of a different evolution of ice sheet topography, the two simulations show a different SMB pattern, leading to a few locations in which the 1-way simulation shows less SMB reduction.





**Figure A2.** Lapse rate of **(a-c)** near-surface (2 m) air temperature [K km$^{-1}$] and **(d-f)** downwelling shortwave radiation [W m$^{-2}$ km$^{-1}$] for **(a, d)** June, July, August (JJA) mean, **(b, e)** December, January, February (DJF) mean and **(c, f)** annual mean. The black lines depict contour lines for every 500 m elevation of the 2-way simulation. The light blue line represents the ELA of the 2-way simulation. The black cross shows the location of the point shown in Figure 5.



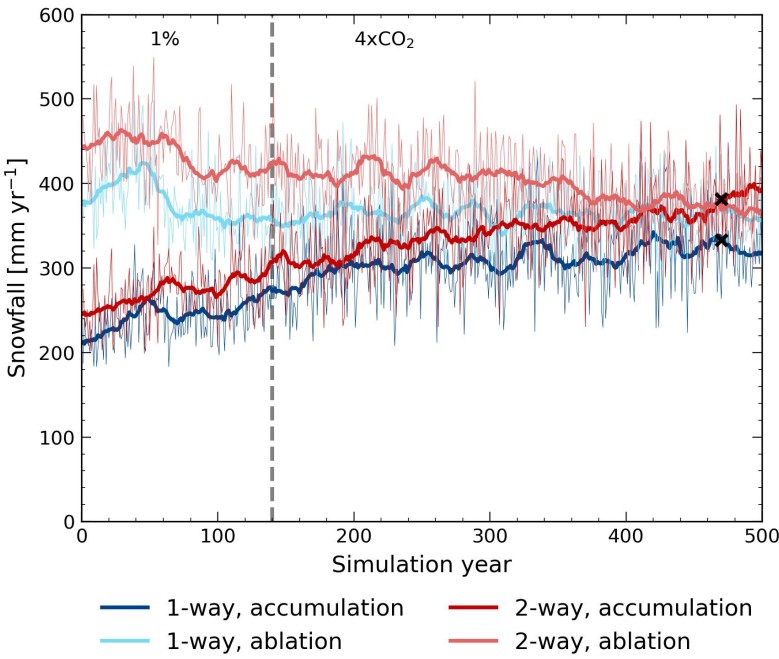

**Figure A3.** Evolution of snowfall [mm yr$^{-1}$] in the 1-way (blue) and 2-way (red) coupled simulation for the end-of-simulation ablation and accumulation area. The grey dashed line indicates the end of the 1% $CO_2$ ramp-up period and the start of the continuous 4x$CO_2$ period. If the differences between the 1- and 2-way coupled simulations become significant throughout the simulation, the first year of significant difference is marked with a black cross.



## Appendix B: $CO_2$ reduction

**Table B1.** Year of recovery. In case the variable has not recovered by the end of the simulation (year 925), the remaining anomaly of the mean state of the variable in the last 100 years of the simulation with respect to the closest boundary of the 1-$\sigma$ confidence interval is computed, as well as the remaining linear trend over the last 100 years. A positive anomaly means that the end-of-simulation state of the variable is larger than in the control simulation. If the trend and remaining anomaly are in the opposite direction, an extrapolated recovery year is given. The relative amount of recovery [%] is given for variables that have not recovered by the end of the simulation.

| | Year of recovery | Remaining anomaly | Remaining trend | Extrapolated year of recovery | Percentage recovered |
|---|---|---|---|---|---|
| Mean global $T_{2m}$ | - | 1.94 K | -0.00022 K yr$^{-1}$ | 9567 | 76% |
| Mean GrIS $T_{2m}$ | - | 2.30 K | -0.0099 K yr$^{-1}$ | 1157 | 68% |
| Mean Arctic $T_{2m}$ | - | 3.71 K | 0.0017 K yr$^{-1}$ | - | 70% |
| Total mass balance | 452 | - | - | - | - |
| Surface mass balance | - | -339 Gt yr$^{-1}$ | -0.23 Gt yr$^{-2}$ | - | 89% |
| Calving flux | - | -444 Gt yr$^{-1}$ | -0.088 Gt yr$^{-2}$ | - | 16% |
| March sea ice extent | - | -1.04 $\cdot 10^6$ km$^2$ | 0.00076 $\cdot 10^6$ km$^2$ yr$^{-1}$ | 2291 | 90% |
| September sea ice extent | - | -5.39 $\cdot 10^6$ km$^2$ | -0.00092 $\cdot 10^6$ km$^2$ yr$^{-1}$ | - | 14% |
| March sea ice thickness | - | -0.64 m | -0.00047 m yr$^{-1}$ | - | 55% |
| September sea ice thickness | - | -0.74 m | -0.000099 m yr$^{-1}$ | - | 9% |
| NAMOC index | - | 4.84 Sv | -0.0062 Sv yr$^{-1}$ | 1707 | - |
| Labrador Sea MLD | - | 115.5 m | 0.80 m yr$^{-1}$ | - | - |
| Irminger Sea MLD | 700 | - | - | - | - |
| Norwegian Sea MLD | 459 | - | - | - | - |
| Nansen Basin MLD | 915 | - | - | - | - |
| Refreezing capacity | - | -0.08 | 0.000024 yr$^{-1}$ | 4352 | 62% |





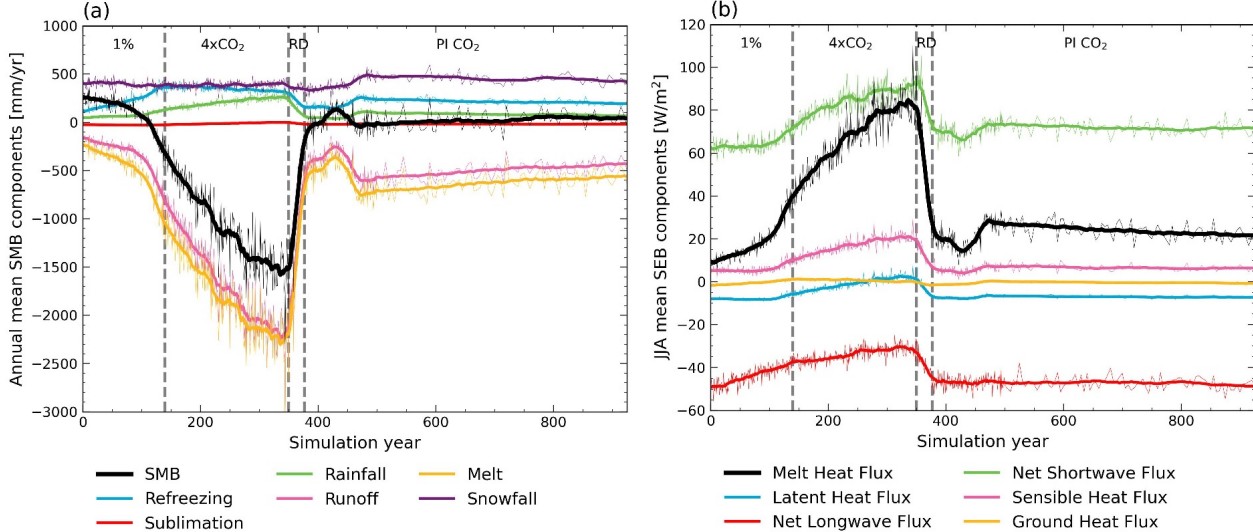

**Figure B1.** Evolution of **(a)** GrIS annual mean SMB [mm yr$^{-1}$] and its components and **(b)** GrIS JJA (June, July, August) mean SEB [W m$^{-2}$] and its components for the simulation in which we reintroduce PI CO$_2$. The grey dashed lines indicate the end of the 1% CO$_2$ ramp-up, the continuous 4xCO$_2$, the ramp-down (RD) to PI CO$_2$ and the continuous PI CO$_2$ periods.



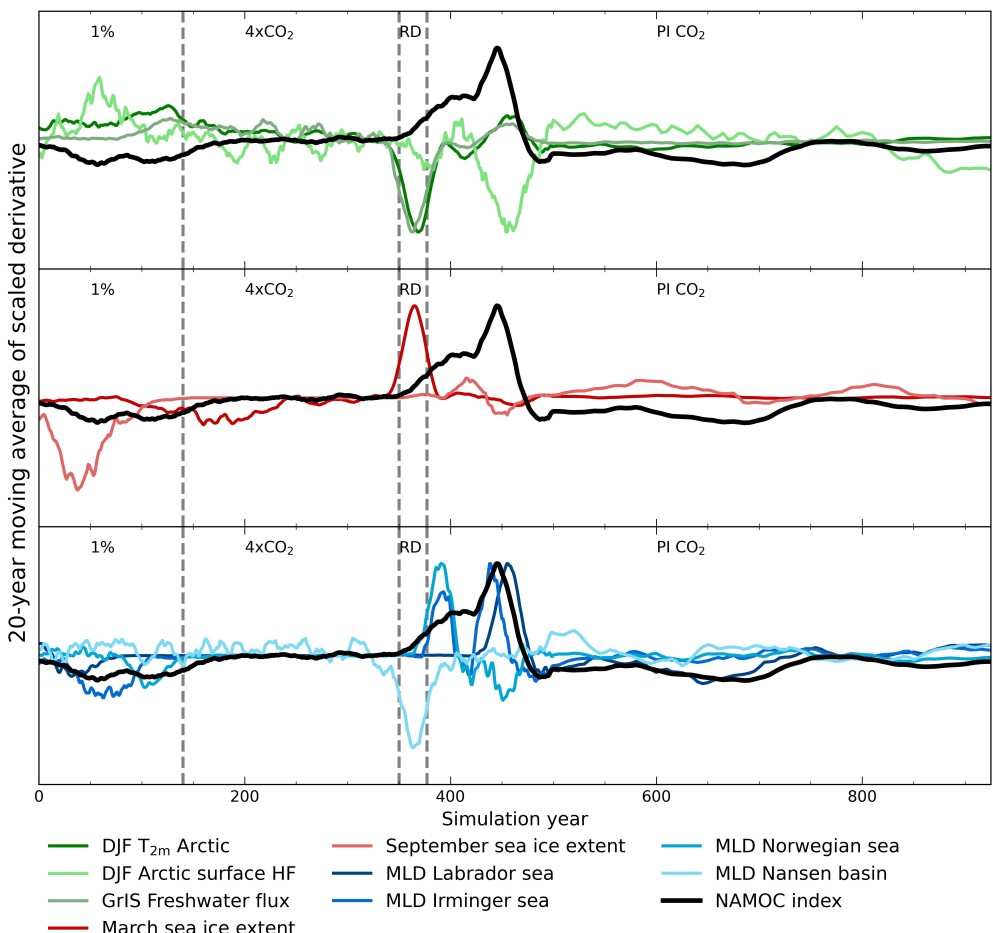

**Figure B2.** 20-year moving average of the derivative of processes interacting with the NAMOC (black). These are the winter Arctic $T_{2m}$, the winter Arctic surface heat flux, the GrIS freshwater flux, the March and September sea ice extent, and the winter MLDs in the Labrador Sea, Irminger Sea, Norwegian Sea, and Nansen Basin. The derivatives are computed using the central difference method and thereafter scaled to their maximum response. The grey dashed lines indicate the end of the 1% $CO_2$ ramp-up, the continuous 4x$CO_2$, the ramp-down (RD) to PI $CO_2$ and the continuous PI $CO_2$ periods.





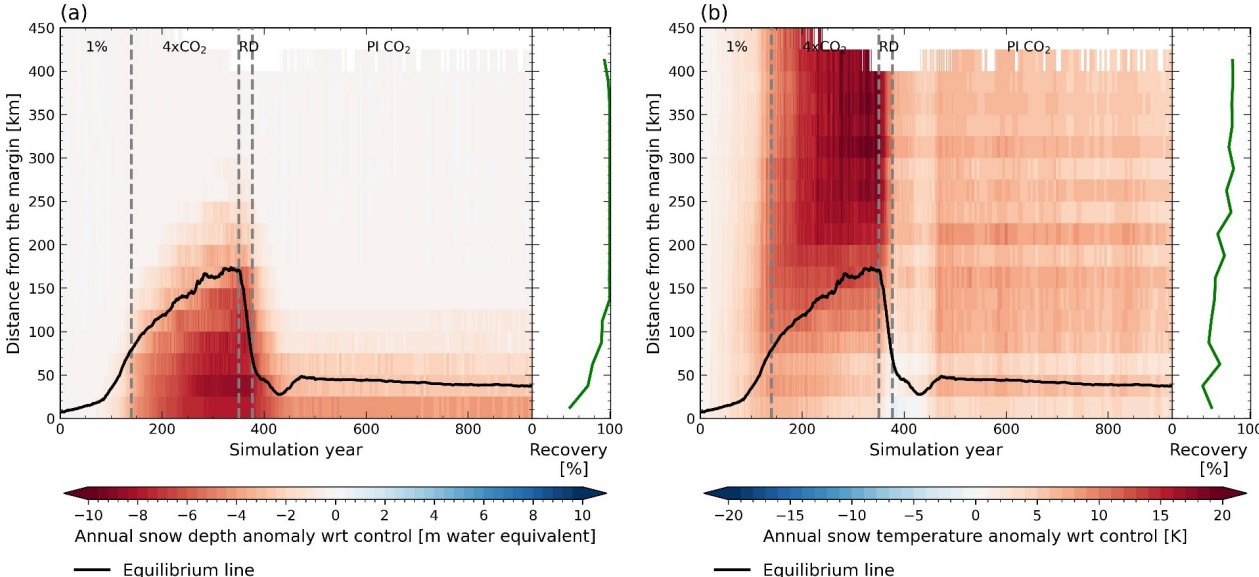

**Figure B3.** Anomalies of **(a)** annual mean snow depth [m water equivalent] and b) the annual mean snow temperature [K] over the GrIS with respect to the control mean as a function of time and distance to the ice sheet margin for the reintroduction of PI $CO_2$ simulation. The black line denotes the mean distance of the equilibrium line to the margin and the dark green line shows the relative recovery (in %) for each distance class. The grey dashed lines indicate the end of the 1% $CO_2$ ramp-up, the continuous 4x$CO_2$, the ramp-down (RD) to PI $CO_2$ and the continuous PI $CO_2$ periods.



*Author contributions.* TF designed and performed the analysis and wrote the manuscript; MV and BW supervised this research; MV, RS,
MP and KTC designed and performed the simulations; all authors read and provided inputs to the manuscript.

*Competing interests.* BW is member of the editorial board of The Cryosphere.

*Acknowledgements.* Leo van Kampenhout is thanked for providing the code to prescribe ice sheet freshwater fluxes for the 1-way simulation.
Sotiria Georgiou is thanked for implementing prescribed freshwater fluxes in the 1-way simulation and providing tools to analyze ocean
model output. MV, MP and RS acknowledge funding from the European Research Council (Grant ERC-StG-678145-CoupledIceClim).
CESM2 is an open-source model, available at https://www.cesm.ucar.edu/. The CESM project is supported primarily by the National Science
Foundation (NSF). This material is based upon work supported by the National Center for Atmospheric Research, which is a major facility
sponsored by the NSF under Cooperative Agreement 1852977. Computing and data storage resources, including the Cheyenne supercomputer
(doi:10.5065/D6RX99HX), were provided by the Computational and Information Systems Laboratory (CISL) at NCAR.





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
