# Peer review of "Role of elevation feedbacks and ice sheet-climate interactions on future Greenland ice sheet melt"

_EGUsphere, 2024_

## Referee Comment (RC1)

**Review of "Effect of elevation feedbacks and climate mitigation on future Greenland ice sheet melt" by Feenstra et al.**

**General impression**

This manuscript builds upon previous work on bidirectional coupling between a dynamic, interactive ice sheet model of Greenland and the relevant components of an Earth system model; in this case, the integration of the Community Ice Sheet Model into the Community Earth System Model, respectively. The technical details of the coupling itself and a range of analyses of the performance of the coupled model for the Greenland domain have been previously presented in a set of papers published mainly between 2020 and 2021. The present manuscript seems to be an extension of that set, filling a gap left by and acknowledged in those studies regarding the actual quantification of the differences –for this particular coupled model– between the bidirectional coupling, a.k.a "2-way", and a simpler unidirectional one, a.k.a. "1-way", more representative of the trends in previous research at the time. In the latter, the dynamic ice sheet is still forced with an evolving climate from the other Earth system model components, but (in contrast to the 2-way coupling) these components are not informed about the resulting ice sheet changes. Thus, 2-way simulations can explicitly represent feedbacks between ice sheet and climate. Long story short, the authors wonder about the causes and significance of the potential differences between 2- and 1-way simulations.

Although 1-way simulations can be performed "offline" by first running the Earth system model with a fixed ice sheet (i.e. no dynamic ice sheet model) and then using its time-dependent climate output to force stand-alone ice sheet simulations, the authors take advantage of the rather flexible coupled setup that seems to allow for specific interactions to be switched off. In this case, it is the response of the climate to changes in ice-sheet surface elevation, while other interactions (e.g. albedo changes) are still accounted for. Thus, two quasi-equivalent simulations can be directly compared, where the effects of climate-elevation feedbacks can be, in principle, precisely isolated. This itself is already a very interesting option which –assuming other interactions mentioned in the text (e.g. albedo-melt, discharge-melt) can also be toggled-- presents an opportunity to assess the contribution, significance, and dominance of individual processes. So far, I think that such an analysis would be a great contribution to the community and enough material for a publication.

Taking advantage of the climatic context of the experiments analysed in this manuscript, comprising a rapidly warming world where pre-industrial atmospheric carbon dioxide levels are smoothly increased four-fold over the first 140 simulation years, the authors add a second part to this study. In what feels like a left turn from the assessment of the differences between 1- and 2-way simulations, this second part looks at the ice sheet and climate response to a relatively sudden reduction of the increased atmospheric carbon dioxide back to pre-industrial levels over 27 simulation years. This new simulation branches out from the year 350 of the 2-way simulation, keeping the fully coupled setup active throughout. The 1-way setup is not mentioned again during this analysis. I have mixed feelings regarding the contribution of this second part to the clarity, conciseness, and completeness of the manuscript. My concerns are elaborated on in the comments below.

**Main points**

1. On the scope of the manuscript

As mentioned above, I am not convinced that the addition of the idealised mitigation analysis (in its current form) adds a positive net value to the manuscript. After reading the first 15 pages of the preprint I, as a reader, felt already well immersed on the quantification of the differences between the performances of the uni- and bi-directional coupling approaches under this idealised warming scenario. I was surprised when I found no mention of a 1-way simulation in the climate mitigation sections, mainly because I was curious about the magnitude difference relative to the 2-way simulation under such a setup. Something in the line of, e.g., a branch-out from the 1-way simulation at year 350 that follows the same mitigation protocol, a version of the 2-way branch-out presented where the elevation feedback is turned off, or both. Maybe, I thought, the need for a 2-way coupling –and the added complexity it brings-- is not that clear in a mitigation context and, depending perhaps on the chosen timing and rate of the forcing reversal, the simpler 1-way coupling setup might do a good enough job. It is possible that an expert might rule this as unlikely based on similar studies or even their expert judgment, but I expected to see numbers; after all, quantification is a core theme in this study. In the introduction, the authors themselves seem to describe this as the original point of the experiment: "to assess the impact of ice sheet-climate interactions […] to $CO_2$ reduction", which happens to be the same wording used in that same paragraph regarding the first half of the study. A bit earlier in the introduction (lines 60-61), the authors mention, without directly providing references, that it is important to account for feedbacks during mitigation studies; well, this paper could be that reference! As it stands now, this manuscript gives the impression of two related but not well connected topics: 1) a quantification of the impact of the elevation feedback, and 2) an analysis of a simulation that mimics 1 potential warming mitigation scenario. Together, the inclusion of these 2 topics affects the conciseness of the paper and consumes resources that could be used instead to fill some of the gaps in either of them (see other comments below for examples of gaps in the former). All in all, I think that the mitigation analysis can be safely removed to make the paper shorter and consistent. I would be happy to see a different manuscript in which the impact of the bidirectional coupling and the hypothesis that "[…] the timing and the rate of the $CO_2$ ramp-up and ramp-down likely have a large influence […]" (line 429) can be properly evaluated and quantified. Then, the title of the present preprint can be easily updated to something more precise, e.g., "A quantification of elevation feedback effects on Greenland melt under an overshoot warming scenario as simulated by the coupled CESM2-CISM2". In this way, the authors can focus on a single topic that is already interesting enough and comes with nice new findings and confirmations regarding precipitation, air circulation, and cloud feedbacks. Just to be clear, I personally would not support the publication of this manuscript without either an inclusion of 1-way simulations in the mitigation analysis (not preferred), or leaving that second part out (preferred).

2. On the significance of the quantification

The authors find that the "1-way configuration overestimates these [topography-related] feedbacks" (lines 468-469), partially point out to the utilisation of a constant temperature lapse correction when downscaling the climate forcing onto the ice sheet model grid, and then hypothesise that "taking this seasonal dependency [of lapse rates] into account […] could help resolve part of the discrepancy between the 1- and 2-way coupled simulations." (lines 473-475). Later, the authors

stress the importance of using bidirectional coupling for projections of future sea level rise (lines 480-481). Now, don't get me wrong; I agree with the core message there. However, I can't help noticing a couple of gaps and leaps in that chain of statements. In this study, the 1-way coupling (i.e. the communication between the climate components and the ice sheet model) is made possible by the temperature lapse rate correction, which seems to do a good job within the context of the elevation classes, but clearly struggles when forced to also cover for the significant departures in ice sheet topography as seen by different model components. In my opinion, presenting a quantification of elevation feedbacks using *only* an homogeneous lapse rate for the latter correction is insufficient to support a call to invest in fully bidirectional coupling setups right away. The authors are aware of work such as Crow et al., (2024), and discuss the possibility that "offline corrections in ice sheet models can be improved by accounting for the temporal and spatial variability of the lapse rate" (lines 424-425). Suppose an ice sheet modeller would like to improve their 1-way projections of sea level rise, but they are not sure about investing in a bidirectional coupling setup. In this case, knowing that a spatially variable lapse rate correction at the core of their 1-way simulations could compensate for a good amount of the differences relative to the not-there-yet 2-way setup --at a comparatively small implementation price-- would surely sound like an excellent option. This study could be that quantitative confirmation if it included that analysis. This way, the quantification presented on this work would get some new context regarding their dependence on an arguably oversimplified approach to very large elevation corrections, shedding some light on how much (or how less significant) the impact is when using a "better flavour" of 1-way. Such addition to this paper would be, in my view, *much* more useful and in-tone with the initial topic of the paper than the mitigation experiments. I believe that the gain in completeness would more than compensate for any apparent loss in novelty that the removal of the mitigation analysis might generate.

3. On manuscript presentation

Although I personally find the quality of the fundamentals (i.e. language and figures) on the high side, I must mention the overuse of acronyms. This seems to be standard practice nowadays, sadly, and at points it makes reading (and reviewing!) a rather painful experience. I imagine it being a bit frustrating for a good-willed general reader that might want to know more about model-based topics in the larger climate change field. I had to go back and forth many times to try and find the meaning of some of them in the text, and the fact that there isn't a summary of acronyms in the shape of a list or table doesn't help. In fact, I think that if such a summary is ever needed, you prolly using too many. Anyway, my point is that a good chunk of the acronyms is likely unnecessary, and to kind of prove it I have so far avoided them altogether in this review. For example, as long as it is clear that your study concerns the Greenland ice sheet, you can safely refer to it as "the ice sheet" or "Greenland" along with "the Greenland ice sheet", which as a bonus add some variation to otherwise tiring/repetitive language choices. If not, a submission to acronyms even ends up creating mildly weird constructions such as "in future sea level rise to the GrIS and the Antarctic ice sheet", which could have used a much more pleasing "the Greenland and Antarctic ice sheets". Similar story with the model components within CESM: other land, atmosphere, ocean, ..., models are seldom mentioned (if ever). Again, you can safely say "our land model" instead of a later and sudden "CLM5" that I then have to search back for. And to be honest, two thirds of typical model names rarely give you any additional information on the nature of the model anyway (particularly true for the Community Models, if I may). Additionally, acronyms can create unnecessary

confusion, such in the use of NAMOC instead of AMOC. There I am no expert for sure, but since you still end up using the latter near the end of the paper I stopped to wonder the need for that N, and whether it was worth it to risk a casual misreading for the Northwest Atlantic Mid-Ocean Channel. I have several other examples, but I think and hope I've made my point. Besides acronyms, I think the paper would greatly benefit from a better and smoother connection between the sentences and some work on word choices. For example, at the conclusions I got heavily distracted with the repetitive use of the word "result" in its many uses and conjugations. Another example is the text in lines 341-344, which repeats the phrase "at the end of the simulation" at least 3 times, even twice within the same sentence. I just want to stress that presentation matters, and that this is a low-hanging fruit that can easily elevate the quality of the manuscript: less acronyms (nothing against them per se), more fluidity in the sentences, use of synonyms, a couple of passes with a native speaker (or a certain tech breakthrough for inspiration), ...; all at a negligible cost compared to (re-)running Earth system model simulations.

**Minor points**

Abstract: connected to my main point above regarding acronyms. Some (e.g. CESM-CISM) are not even used within abstract again (and reintroduced later in the text, so no difference). On that note, if the title is changed to include the acronym of the model, then there's no loss of info at all. Same with using a construct twice within a same sentence (e.g. "losing mass […] lose mass"). To exemplify my point, here's a humble suggestion for the first half of the abstract:

"The Greenland ice sheet stores freshwater equivalent to more than seven meters of potential sea level rise and strongly interacts with the regional and global climate. Over the last decades, the ice sheet has been losing mass at a rate that is projected to increase in the [some future timescale]. Interactions between the ice sheet and the climate have the potential to amplify or reduce its mass balance response to ongoing and projected warming. Here, we investigate and quantify the impact of these interactions using the Community Ice Sheet Model version 2 coupled to the Community Earth System Model version 2. We compare two idealized simulations --containing either a non-evolving or evolving ice sheet topography-- in which we apply an annual 1% increase in $CO_2$ concentrations until stabilization at four times pre-industrial levels. By comparing the 1- and 2-way coupled simulations, we find that applying a uniform temperature lapse rate to account for elevation differences among model components in our 1-way experiment misrepresents temperature changes in the ablation area, leading to an overestimation of the positive melt-elevation feedback and the resulting mass loss. Furthermore, our analysis reveals significant changes in atmospheric blocking, precipitation and cloud formation over Greenland as the ice sheet topography evolves, which act as important negative feedbacks on mass loss. [...]"

I'll spare the authors of more comments about acronyms and words, unless necessary, but my view stands for the whole manuscript.

Line 50: "overshoot" is introduced in quotes, but a definition of overshoot scenarios is not given. Consider adding it for the general reader, e.g. "[…] investigation of 'overshoot' scenarios, where this temperature threshold is surpassed […]".

Line 50: " Applying a temperature overshoot to the GrIS" is not precise language, consider something like "A climate where global mean temperatures have increased beyond the 1.5°C goal might have large implications for […]". Or even better, and after defining 'overshoot', simply stating "Such an overshot could have important implications […]".

Line 52: which period?

Line 54: feasibility for what? For a recovery under a ramp-down?

Lines 56-57: Is that sentence (which in essence repeats the info from the previous one) also a finding of the studies cited? The next sentence implies that it is a conclusion drawn from modelling studies. If yes, consider rephrasing it to reflect that, e.g. "Model-based results from these studies seem to suggest that if such thresholds are not crossed, ice sheet retreat can be halted or might even be reversible.", which by the way highlights my observation about repeated info and provides a bridge to the next sentence.

Lines 57-59: Sounds weird. Consider something like "However, a thorough model-based assessment of the role played by ice sheet-climate feedbacks in the reversibility of enhanced deglaciation rates is currently lacking."

Lines 62-67: This paragraph sounds very model-specific for an introduction section, and ignores existing research with other models (e.g. Madsen et al. 2022). It can be easily rephrased to coupled setups in a general sense and acknowledge other groups worldwide working with bidirectional coupling, plus some context. As a bonus, this would solve the issue of introducing the models (and acronyms) twice in the manuscript. CESM-CISM-specific sentences can be moved to the model description below. The remaining 2 mentions of CESM-CISM can be replaced by "a coupled ice sheet and Earth system model" and removed, respectively.

Line 140: The first half of the simulation design section is just a description of the 1w coupling. I think this could go into the coupling section, which can be divided into 2-w and 1-w for clarity. If the design of 2-w is taken 1:1 from previous studies, I'd like to see it made clear in the text; since at the moment it is somewhat implicit but still unclear. In other words, if any design or simulations are taken directly from previous work, I'd appreciate if there is a clear separation from what is brand new in this study.

Line 143: Does this mean that the freshwater fluxes are fixed and the discharge-melt feedback also removed from the 1-w? If yes, why so? So far the text has given the impression that only the elevation-melt feedback is targeted for analysis. Then why not turning off the albedo-melt feedback as well? I would like the authors to elaborate on the reasoning (or practicalities) behind these choices.

Line 156: kept constant for how long? Would be nice to have the total length of the simulation here as well.

Lines 230-231: Wouldn't it make more sense to compute then the percentage of the continent/island/whatever that experiences ablation?

Line 467: 66% more melt than what? 1-way? PI melt? Please specify. Same with other percentages elsewhere.

Line 471: missing "/km" in the units of the rate?

Line 472: what do you mean here? You mean the "real" rate?

**References**

Crow, B. R., Tarasov, L., Schulz, M., and Prange, M.: Uncertainties originating from GCM downscaling and bias correction with application to the MIS-11c Greenland Ice Sheet, Clim. Past, 20, 281–296, https://doi.org/10.5194/cp-20-281-2024, 2024.

Madsen, M.S., Yang, S., Aðalgeirsdóttir, G. et al. The role of an interactive Greenland ice sheet in the coupled climate-ice sheet model EC-Earth-PISM. Clim Dyn 59, 1189–1211 (2022). https://doi.org/10.1007/s00382-022-06184-6
* * *
Jorge Bernales, June 2024

---

## Referee Comment (RC2)

**General comment**
The authors propose an evaluation of the effect of the inclusion of a changing ice sheet geometry on the climate around Greenland and vice-versa. They do this in part 1 of the manuscript by comparing 2 simulations carried out with the Earth System Model CESM2 coupled to the ice sheet model CISM2, in a ramp-up scenario where major parts of the ice sheet are lost.
The authors mention that, despite that 2-way coupled climate and ice sheet simulations of the GrIS had already been carried out with CESM2-CISM2, there had never been an evaluation of what the effects of including a 2-way coupling in CESM2-CISM2 would be. They also mention that this is the first time the effects of GrIS topographic changes on blocking events was being investigated, despite it being mentioned in the literature for some time now. I found it really interesting that the effect of changes in topography on blocking and the other negative feedbacks outweigh the melt-elevation and melt-albedo feedbacks, as those are often mentioned first as a justification for using coupled climate and ice sheet models. In that way, this manuscript brings something new and much needed.

The manuscript is well written and generally easy to read and my comments mostly concern rewriting some sentences to make the reading a bit easier. The authors were also really thorough in their justifications.

**Comment on the second part of the manuscript**
I do however have an issue with the second part of the manuscript, the one where the ramp down scenario and the investigation of the effect of a cooling on the GrIS is introduced. I don't think it really belongs with the first part, as this one is really an evaluation of an "improvement" in a model. This, along with a few other things, made me think the manuscript was a (condensed version of) a student's thesis. A quick check online confirmed that it was a masters thesis.
In a thesis, there's a common thread where different sections are related to each other but are their own distinct piece of work. Putting them together in one paper doesn't really work in my opinion as, in its current form, the manuscript has 2 messages/questions: the first being "what effect does including a 2-way coupling has on future projections of the GrIS melt?" and the second "are there conditions under which the projected melt of the GrIS can be reversed in CESM?".

Ideally, if you had branched a 1-way ramp down simulation from the same point of the 2-way ramp up as the 2-way ramp down simulation, you could have done a similar comparison as in the first part of the manuscript where you'd have evaluated the effect of a 2-way coupling on the recovery of the ice sheet. But running this extra simulation, if you haven't already, is going to take a long time.

The results and the described processes (NAMOC overshooting, the existence of a smaller but stable ice sheet in the future and the comparison to the PI state at the same temperature ...) are interesting so I am not saying that they shouldn't be published at all. But I am also not entirely convinced that part 2 has enough material to be smaller paper or brief communication in its current form. As you say, "the timing and the rate of the $CO_2$ ramp-up and ramp-down likely have a large influence on the mass balance evolution and influence whether mass loss can be halted or reversed " and you'd probably have to re-write extra bits to justify the work. Have there been other similar simulations done by other CESM2 users you could include as comparison? Or could this be part of a larger study involving, e.g an overshoot scenario study with CESM2-CISM2?

In conclusion, I think that, worse case scenario, part 1 is enough for a paper and turning a masters thesis into a peer reviewed manuscript of this quality is already an achievement. But I don't want to discourage a young researcher from publishing their work either so, if you can't find a way to publish the second part as a separate paper but really manage to justify its inclusion in the present manuscript more (including by modifying the title a bit to change the message maybe?), I will not

be opposed to its publication in its current form.

**Minor comments**

General comment on figures: some figures might benefit from being slightly wider (e.g. maps). I liked the consistent use of colours but the shades of blue, red and green were sometimes difficult to tell apart, especially in figure 2d and again in figures 7b and 9b. Could you keep the same main colours but change the tint/shade (i.e. make darker tones even darker and lighter ones lighter)?

**Abstract**
**p1, line 10:** I suppose the lapse rate you're mentioning is the lapse rate used to downscale temperature from the elevation tiles to the ice sheet grid. It would be useful to add that information.
**p1, lines 12-13:** add that it is for the 2-way coupled simulation

**1. Introduction**
**p2, line 52:** what do you mean by "such a period"? A certain length of time or simply that it's a warmer period?

**2. Method**
**p4, lines 96-97:** I'm not sure I understand these 2 sentences. Is the snowpack thickness reset to 10m at the beginning of every year? Meaning that if the thickness at the end of the year is 10+X m, X m is the positive SMB that is transferred to CISM2? And, in the second sentence, what do you mean by "further melt"?
**p4, line 119:** I don't think the definition of refreezing capacity is necessary here, as you only mention it much later in the manuscript (p19, end of section 5). Instead, I would just change line 376 (p19) in "The refreezing capacity (amount of refreezing divided by the amount of available water) peaks earlier..."
**p5, line 126 (then p6, line 146 and p11, line 252):** The way you wrote these different sentences, I am not sure whether the fixed lapse rate of -6K/km is used in both the 1- and 2-way simulations. From line 126, I think yes but then I was a bit puzzled when reading "*as is done in the 2-way coupled configuration*" in line 252. Finally, in lines 250 to 254, you mention a computed lapse rate that you compare to the fixed lapse rate of the 1-way simulation.

If I understood correctly, the lapse rate used for downscaling the temperature from CESM's to CISM's grid is fixed in both simulations. I'd add in line 126 that it is the case in both simulations, to make it clear the first time it's mentioned. And I'd remove "*as it is done in the 2-way coupled configuration*" entirely in line 126.

Then, in section 4, lines 250-254 I think the lapse rates that you mention you're computing are computed offline, in the same way as you computed the lapse rates for the ME feedback. I'd add here for clarity that those are computed offline I would then add, either in line 252 or 253, something along the line of "*compared them with the fixed applied lapse rate used to downscale temperature and LW during the simulations*". I'd also remove the 1-way simulation mention if the 2-way simulation also uses a fixed lapse rate for temperature and LW. I think adding that would make the reader know immediately which lapse rates you are referring too and would make the reading easier.

**p5, line 137:** You're mentioning the fact that, since the coupling from POP2 to CISM2 is not implemented yet (presumably because you need a way to downscale ocean temperatures onto the ice sheet grid in order to be able to resolve the fjords), there is no direct influence of the ocean on the ice sheet via it's forcings on marine terminating glaciers. Could you add a few words about the potential biases this could lead to and the processes involved?

**p6, line 165 to p7, line 193: Section 2.4 (metrics definitions)**
I don't think this section is necessary in this form. Some of the concepts are useful to define in a thesis but are well known to the readers of scientific papers (e.g. the definition of ELA) and others are only used much later in the manuscript and should, in my opinion be moved there. I'd keep the definitions of lapse rates, GBI and the moving average and remove the ELA, NAO and IVT.

- Lapse rates: I am not entirely sure what the lapse rates refer to here. I think you're using them to isolate the melt-elevation feedback as this would be the only way to evaluate that the ME feedback leads to more melt in the 2-way simulation since the SMB doesn't decrease as much in the 2-way simulation and the snowfall doesn't differ much. I'd move the definition to section 4.1.
- GBI: I'd keep the definition in the manuscript as you're using the modified GBI proposed by Hanna et al. (2018), which they call GB2. I'd move the definition to section 4.2 and would add that what you're using is called GB2 in Hanna et al.
- Moving averages: I'd keep that one to remind the reader that the length of the moving average changes during the simulations but I'd put it at the end of section 2.3 (simulation design).

**3. Simulated mass loss**
**p8, lines 216-218:** Is there a numerical threshold in the NAMOC index for you to consider it to collapse or are you looking at increased rates of change?

**4. Climate feedbacks**
**p12, line 262:** I'd add *orange dashed line* after *Figure 5c* so the reader can spot the point without reading the caption.
**p11, caption figure 4:** the blue line shows the monthly mean surface temperature, not the red line.

**6. Discussion**
**p21, line 425:** *By considering the surface temperature* ? What does considering mean here?

**Typos, spelling, punctuation**

**p4, line 103:** ice thermodynamicS model instead of thermodynamic? Like in ice dynamics model?
**p4, line 106:** same
**p6, line 140:** simulationS design?
**p6, line 146:** -6 K km$^{-1}$ instead of K/km (as p5, line 126)
**p6, line 148:** "In contrast, the 2-way coupled run..". *to the 1-way coupled run* in not necessary here as you were just talking about the 1-way coupled run in the previous sentence.
**p6, line 157:** I'm not sure the last part of the sentence (after *which*) is grammatically correct. I would write "the 4xCO2 scenario is an extreme warming scenario and, after the year 140, has a similar radiative forcing to that of the SSP5-85 scenario at the end of the 21$^{st}$ century".
**p7, lines 197&203+p8, line 206:** 20-year centered moving average (with a hyphen)

**p8, line 207:** returns within one standard deviation OF the PI mean.

**p10, line 244:** no comma after *we account for this*
**p10, line 245:** comma after *while* if you put one after *1-way simulation*
**p12, line 264:** no comma after *simulation*
**p15, line 311:** *not significantly* instead of *not significant*

**p16, line 335:** here you express SLR in cm but in line 321 you express it in m. Can you check

throughout the manuscript and pick one?
**p18, line 372:** no comma between *forcing* and *are*

**p21, line 395:** *as* opposed to?
**p21, line 399:** comma before *although*
**p21, line 403:** comma after *cloud cover*
**p21, line 404:** the sentence is a bit long so I'd start a new one after *transmissivity* (This aligns...)

**p22, line 433:** *tens of thousands of years*
**p22, line 448:** no comma after *level*
**p22, line 449:** no comma after *small*
**p22, line 450:** *if we were extending* I think

---

## Referee Comment (RC3)

A Review of: "*Effect of elevation feedbacks and climate mitigation on future Greenland ice sheet melt*" Feenstra *et al.* 2024, The Cryosphere Discussion.

Feenstra et al. propose a study about the different feedbacks related to the interaction of the near-surface atmosphere of the Greenland ice sheet with its evolving topography following $CO_2$ concentration varying in the atmosphere. To represent these interactions, they use the model CESM2 coupled with the ice sheet model CISM2. The study is divided into two main parts depending on the analyzed experiments.

The first part concerns a comparison between 1-way and 2-way coupling method, and more specifically an evaluation of the non-consideration of feedbacks related to the evolving topography of the ice sheet. Components of the mass balance and the surface mass balance are compared, as well as the influence on the GBI and cloud formation. They highlight different negative feedbacks linked to the evolving surface topography that mitigate the mass loss in a climate with 4x the $CO_2$ concentrations of the pre-industrial period.

Next to that, the extent of the 2-way experiment is realized by rapidly decreasing the $CO_2$ concentration until reaching the PI concentration. Before the decrease of $CO_2$, the ice sheet had experienced 350 years of a climate with 4x $CO_2$ concentrations, resulting in a retreated ice sheet and a largely reduced ice discharge. Combined with a decreased melt at the surface due to a colder climate, the ice sheet presents a limited ice loss, and even a slightly positive SMB when the climate reaches a global warming of 2 K. The oceanic and snowpack conditions limit the possibility for the ice sheet to regrowth under the PI-reconstructed climate.

*General comments*

This manuscript is well-written and of generally good quality. Scientific approaches are pertinent and consistent. The authors propose a study of a wide range of the topography influences around the Greenland ice sheet, which constitutes both its strength and its weakness. I will have four major comments.

First, on one hand, the study is well complete and treats a lot of possible influences resulting from the interactions between the atmosphere and evolving surface height of Greenland (comparison 1- and 2-way method of coupling, GBI, NAMOC, cloud formation, energy balance and idealized PI $CO_2$ concentrations restoring and its influences). But on the other hand, all the topics and various processes studied are not deeply analyzed. Moreover, the part of the study concerning the decrease in $CO_2$ concentrations experiment answers a different question than the part concerning the comparison of 1-way and 2-way coupling. You present in the paper various messages, which could be, to my point of view, better highlighted if they are split. I suggest then dividing this manuscript

into two distinct papers: a) 1- and 2-way experiments study, and b) the study of the idealized PI $CO_2$ concentrations restoring experiment. This way, you could dig a bit deeper into each topic, without presenting a too long and dense manuscript with several messages.

Concerning one of the topics that you should revise, I would like to highlight in a general comment that it seems you miss one thing concerning the evolution of the GBI with the 2-way coupling. As the surface height is lowering with time, the 500 hPa geopotential height is also lowering independently of any atmospheric circulation modification. Even if the index removes the 500 hPa GH of a larger area than the Greenland GBI area to the 500 hPa GH of Greenland, I think the simple lowering surface height have a non-neglectable influence that you should consider in your analysis. I comment on this point in the specific comments.

Finally, I have two comments concerning the methodology.

It seems that the description of the coupling and some details about the models are missing to well understand how the experiments are set up independently of the paper describing the coupling. I wrote specific comments in the next part of the review that you should address to improve this part of your study.

I'm a bit confused about the motivation of the 1- and 2-way simulation comparison, as well as the use of a fixed temperature lapse rate. I'm a bit annoyed by the fact that you explain right from the introduction that using a fixed lapse rate is not at all the best way to represent melt-elevation feedback since this lapse rate varies spatially and in time, and is influenced by these different feedbacks linked to topography. Despite this context, similar and more detailed arguments are repeated in the discussion to suggest that it would be more appropriate to use a non-fixed lapse rate. If I understand well, it seems that you had to use this fixed lapse rate to respect the methodology already used by the coupling and two compare things that are comparable. The fixed lapse rate also helps to highlight processes that are not considered when compared with 2-way coupling. However, none of these justifications appear clearly in the paper. In the introduction, you only mention the need to quantify the incorporation of such a coupling into CESM2 (P3 L67-68). But in the conclusion, you mention the improvement of the 1-way VS 2-way comparison (L 474: "[...] could help to resolve part of the discrepancy between the 1- and 2-way coupled simulations"), which may lead to confusion regarding the motivation of your comparison and the use of such a 1-way method. Especially as various other more efficient methods for representing melt-elevation feedback through offline downscaling of the temperature (Hanna et al., 2005; Gardner et al., 2009; Crow et al., 2024 that you mentioned), or directly from the SMB (for instance: Noël et al., 2016; Goelzer et al., 2020 for ISMIP6; Delhasse et al., 2024) have already proved their effectiveness. Therefore, it would be a good idea to adapt the message to clearly highlight the reasons for such 1-way and 2-way comparisons and the use of this fixed lapse rate.

_Specific comments_

**Abstract**

- P1. L6-8: Please add a time reference for your simulations and specify the level of $CO_2$ at which you start.
- P1, L10-12: "We also find that a uniform temperature lapse rate misrepresents temperature changes in the ablation area, leading to an overestimation of the positive melt-elevation feedback in the 1-way coupled simulation, resulting in an overestimation of mass loss." I think it's inappropriate to consider that the overestimation of the melt-elevation feedback by the 1-way simulation is an original and new result of your study as you already mention in your introduction (P2 L41-43) and in your discussion that other different studies with similar one-way experiments have already the same conclusion. You should rephrase and nuance this sentence.
- P1 L12-13: precise that you extend your 2-way coupled simulation instead of a new experiment.

**Method**

- P4. L94-99 "The model has a fixed number of vertical layers for the soil, whereas there is a variable number of layers for snow and firn, with a maximum snow depth of 10 m water equivalent (w.e.). The model allows for compaction of snow into firn. Accumulation of snow over 10 m w.e. in a grid cell is transferred as positive SMB to CISM2. If snow and firn are melted away, further melt is transferred as negative SMB (ice ablation) to CISM2." Does it mean that having less than 10 m w.e. of accumulation is not considered in the SMB ? Same for ablation?
- P4, L119: Please add equations for SMB and refreeze capacity to illustrate and summarize how SMB and refreeze are considered.
- P4: In general, has your model and specifically the representation of the SMB, the key feature of your coupling experiment already been evaluated against observation? How robust are your results? What are the range of bias for the different processes you are representing?
- P5 L126 & L.145: "The resulting climate and SMB are downscaled using elevation classes (Sellevold et al., 2019), using a temperature lapse rate of -6 K/km and are interpolated onto the CISM grid". The way you explain how the temperature lapse rate is used to interpolated the SMB is not clear enough for me. Are the SEB and SMB also downscaled with a lapse rate? Or are they calculated on the CISM grid after the temperature has been interpolated using the constant lapse rate? This part of your method should clarify as this is the only way you consider the melt-elevation feedback in your 1-way coupling experiment. Furthermore, could you briefly explain the elevation class method?
- P6, L142-143: How the initial topography of the ice sheet is retrieved? Is it the usual topography used by CAM6? Or is there a step of initialization for CISM? Same questions for freshwater fluxes, from where are they come from? Also, if you use a

different topography than the usual one for CAM6, how the atmospheric module is answering to, sometimes, big differences in elevations?

- P6 L.152-153: "In contrast to the surface topography, the surface albedo is updated as a response to ice sheet melting in both the 1-way and 2-way coupled simulations." I had to read several times this explanation to really understand what it means. I guess then this is not the best way to explain it. You should more deeply explain what implies this consideration of the albedo. Also, as in the 2-way simulation the ice sheet is retreating, is the albedo-feedback not mitigated by the smaller area of higher albedo (ablation area)?
- P6 L.155: Please specify here how much time your experiments are running. For more clarity, I suggest also to refer to figures 2a and 9a to illustrate you experiment designs.
- P6 L164: Concerning your control simulation, why only have one control simulation with 2-way coupling method? Did you compare it with a similar experiment but with the 1-way method? In other words, does the method to represent the melt-elevation feedback influence the control simulation even if the melt-elevation feedback is supposed to be weak with this CO2 level?
- P7 L179-180 and 183: Could you specify how you normalize with respect to the control simulation? Is it mean you're considering the variability of the GBI/NAO from the control run to normalize the GBI/NAO from 1w- and 2w-experiment?
- P7 L190-193: Usually, a period of 30 years is considered to talk about climatology mean. Is the choice of 20 years/data point influence your results compared to a 30-year average?

**Results**

- Fig 3c: The difference between a) and b) gives still a rate per year. This is not sound to the topography feedback to me. I suggest also illustrating the final result in terms of topography with the differences in meters between topography of 1-way and 2-way for year 500.
- P11, L251: Please justify why you also look at the LW down (and not LWup, SWdown, or up).
- P12 L 273-275: I suggest at least adding these references to describe atmospheric blocking: Hanna et al. 2014, McLeod and Mote 2016.
- P12 L276: Are the differences in GBI between both simulations not simply linked to change in height of the surface, at least partly, thus decreasing the geopotential height of 500hPa? If your surface is lowering, the height of the 500hPa geopotential is also lowering, especially as you have differences in surface elevation up to 1000m after 500 years of coupling. And then this GBI decrease will not be entirely due to "real" changes in blocking event regime, and more generally changes in larger scale circulation. Also, I'm surprise to have such differences, even just in winter, in GBI, and not correlated at all with differences in NAO, as these 2 indexes are partly anti-correlated for the current period (Hanna et al., 2015).
- P13 L 281-292: In this part of your results, you should consider the influence of altitude on the GBI computation (as explained in the former comment) before reaching any

conclusions on the relationship between melting and blocking events. Same comment for the discussion (P21 L 406-415) even if this part is already well nuanced.

- P 16 L 315-317: Despite the accuracy of the explanation concerning precipitation, this analysis could be mitigated by comparing the relative importance of the feedbacks mentioned, compared with melt-elevation feedback (Fig 3 and Fig A1h VS. Fig 8) as precipitation is a much lower contributor to the differences between 1- and 2-way experiments.

- P16 L345: "However, the retreated ice sheet margins result in a smaller contribution of ice discharge to the mass balance (Figure 9d)." If I'm right, could you specify that you compared to the PI situation in this sentence? Also, when you consider the "first" state of your comparison (Table 2), please, indicate the years, to be clearer.

- In this comparison (Table 2), you should insist on the fact that, by recovering the same global temperature anomaly, the state of the ice sheet is quite different, as well as the components engaged in the total mass balance. It could be also interesting for your analysis to have a spatial representation of the ice sheet extent for these 2 specific states, and more generally to illustrate what becomes the ice sheet after such a decrease in $CO_2$.

**Discussion**

- P21 L412 : "However, Hanna et al. (2018)", Add Delhasse et al 2021, which is the updated version of Hanna et al. 2018 with CMIP6 models.
- P22 L432: Please specify that you're mentioning 1.1m of SLR contribution.

**Typo**

- P2 L51: Please define GMSL;
- L672: The reference of Sellevold *et al.* 2019 should be updated as the paper is not in discussion anymore.

**References:**

Delhasse A, Hanna E, Kittel C, Fettweis X. Brief communication: CMIP6 does not suggest any atmospheric blocking increase in summer over Greenland by 2100. Int J Climatol. 2021;1–8. https://doi.org/10.1002/ joc.6977

Delhasse, A., Beckmann, J., Kittel, C., and Fettweis, X.: Coupling MAR (Modèle Atmosphérique Régional) with PISM (Parallel Ice Sheet Model) mitigates the positive melt–elevation feedback, The Cryosphere, 18, 633–651, https://doi.org/10.5194/tc-18-633-2024, 2024.

Hanna, E., Fettweis, X., Mernild, S.H., Cappelen, J., Ribergaard, M. H., Shuman, C.A., Steffen, K., Wood, L. and Mote, T.L. (2014) Atmospheric and oceanic climate forcing of the

exceptional Greenland ice sheet surface melt in summer 2012. International Journal of Climatology, 34, 1022–1037. https://doi.org/10.1002/ joc.3743.

Hanna, E., Cropper, T.E., Jones, P.D., Scaife, A.A. and Allan, R. (2015) Recent seasonal asymmetric changes in the NAO (a marked summer decline and increased winter variability) and associated changes in the AO and Greenland blocking index. International Journal of Climatology, 35, 2540–2554. https://doi.org/10.1002/joc.4157.

McLeod, J.T. and Mote, T.L. (2016) Linking interannual variability in extreme Greenland blocking episodes to the recent increase in summer melting across the Greenland ice sheet. International Journal of Climatology, 36, 1484–1499. https://doi.org/10.1002/joc.4440.

Noël, B., van de Berg, W. J., Machguth, H., Lhermitte, S., Howat, I., Fettweis, X., and van den Broeke, M. R.: A daily, 1 km resolution data set of downscaled Greenland ice sheet surface mass balance (1958–2015), The Cryosphere, 10, 2361–2377, https://doi.org/10.5194/tc-10-2361-2016, 2016.

---

## Author Comment (AC1)

We thank the reviewers for their constructive feedback on the manuscript. In the following, we provide a short joined response to all reviewers. Thereafter, a response to the specific comments of reviewer 1 is given.

**Joint response to all reviewers**

*On the scope of the manuscript and request to add one-way coupled simulation for the 4x to 1xCO2 reduction scenario* : the scope of the manuscript is to examine Greenland ice sheet and climate **interactions**. Feedbacks are one specific type of these interactions, namely those that involve a bi-directional coupling (initial process is augmented or reduced through the feedback). We will make the interaction-feedback distinction more explicit in the introduction of the reviewed manuscript. Quantification of the albedo feedback for 4xCO2 has been done in previous work (Muntjewerf et al, 2020) by examining the contribution of absorbed solar radiation to the total melt energy and a dedicated simulation is not necessary. For this reason, here we focus on the elevation feedback. Since elevation does not change in the mitigation scenario (mass balance becomes approximately zero), we find that it is unnecessary to explore elevation feedbacks there with a one-way coupled simulation.

In addition, we want to clarify that the primary goal of the manuscript is ***not*** to quantify the difference in melt projections for ice-sheet-only and coupled models. We do this only for our model, and the results will be different for other climate models and surface mass balance schemes. In our paper, this numerical comparison makes one part of the manuscript, with the main focus being the physical ***processes*** of ice sheet and climate interaction, and how our model represents them in the one-way and two-way coupled flavors. We will make this more explicit in the reviewed manuscript.

*Suggestion to run more simulations*: Here we present a set of multi-century "IPCC-type" Earth System Model simulations with a 1 degree atmosphere and dynamical ocean components. This type of model is extremely complex and simulations are computationally very expensive (3,600 core hours are required to run one simulation year). To our knowledge, here we are presenting the first comparison of one-way to two-way simulation with an IPCC-type model. In addition, we present the first assessment of the coupling of global climate, ocean circulation and GrIS snow/firn evolution with an IPCC-type model for a scenario of mitigation.  We don't have the means to run more simulations.

*Suggestion to eliminate or move the $CO_2$ reduction simulation to a different paper for consistency or to highlight results separately*: we consider this unnecessary as the common theme here is the assessment of processes of ice sheet and climate interaction. The current structure of this manuscript around the theme of ice sheet-climate interactions first shows the effect of elevation feedbacks by looking at an extreme warming scenario and comparing a set-up with and without evolving GrIS topography, and thereafter addresses other interactions (ocean, snow pack) in the light of a mitigation scenario, aiming to quantify the effect of different interactions and feedback on the GrIS mass balance. Besides, the use of different simulations to address one research question (In our case: "Which interactions between the GrIS and the climate affect the GrIS mass balance?") is not uncommon (e.g., see Gregory et al. (2020), analyzing one 1-way and

several 2-way coupled simulations for different warming scenarios and for multiple mitigation scenarios, around the theme of irreversible mass loss). We propose to make some changes to emphasize more on the common theme in this manuscript (interactions and feedbacks) and the connection between both parts.

To make the common theme clearer we propose to change the title to: "Role of elevation feedbacks and ice-climate interactions on future Greenland melt"

*Request to run more simulations to provide a "one-fits-all" seasonally varying lapse rate for one-way simulations*: we believe this lapse rate will depend both on the modeler choice of climate model forcing and surface mass balance calculation. In this manuscript we do provide a seasonally varying estimate of the temperature lapse rate by comparison of two-way and one-way simulations in CESM. To our knowledge, nobody has provided this sort of estimate. We expect estimates from other models to follow. Crow et al. (2024) is a different type of assessment, where they try different prescribed lapse rates and see which one/type results in a better fit to proxy records.

*Request to clarify one-way simulation design:* the one-way simulation has evolving albedo as this is calculated interactively in the land component. Ice sheet area and elevation are not evolving in the climate components. Meltwater fluxes to the ocean are not evolving. They are prescribed to those calculated in the pre-industrial simulation. We will clarify the simulation design (choices) further in the reviewed manuscript.

*Request to provide justification of fixed lapse rate choice in one-way simulation*: a fixed lapse rate was chosen for consistency with the standard design for sub-grid surface mass balance simulation (downscaling) through elevation classes. Other state-of-the-art downscaling techniques suggested by reviewer 3 are not applicable to an Earth System Model as they are based on high-resolution regional modelling at the scale of 10 km.

*Questions about albedo feedback*: the albedo feedback has been already quantified in a previous study (Muntjewerf et al., 2020). This can be done by looking at the energetic contribution of albedo change (in $W/m^2$) to the total melt energy. That is, there is no need to perform dedicated sensitivity simulations to quantify this feedback. We will make this more explicit in the revised manuscript.

**Response to specific comments of reviewer 1**

*On the use of acronyms and sentence structure:* We agree that "the Greenland and Antarctic ice sheetsyou're your example is more pleasing, however, we feel that the use of the acronym GrIS is so common that this should not be a problem. Regarding our model component acronyms, we will revisit the text and change these to "land/ocean/... model" from section 2.2 forward. Next to that, we will have a look at the connections between the sentences and chosen words, especially for the conclusions and the sentences describing the timing within our simulations.

**Response to minor points that are not answered above**

Referee comments in black, authors' response in red

Line 50: "overshoot" is introduced in quotes, but a definition of overshoot scenarios is not given. Consider adding it for the general reader, e.g. "[...] investigation of 'overshoot' scenarios, where this temperature threshold is surpassed [...]".

Thank you, we will follow this suggestion.

Line 50: " Applying a temperature overshoot to the GrIS" is not precise language, consider something like "A climate where global mean temperatures have increased beyond the 1.5°C goal might have large implications for [...]". Or even better, and after defining 'overshoot', simply stating "Such an overshot could have important implications [...]".

We will include your second suggestion.

Line 52: which period?

We will change this to: "during an overshoot period"

Line 54: feasibility for what? For a recovery under a ramp-down?

Feasibility in the light of policy-making. To answer the question of whether mitigation after a temperature overshoot period can be used to reverse any "damage" that has been done.

Lines 56-57: Is that sentence (which in essence repeats the info from the previous one) also a finding of the studies cited? The next sentence implies that it is a conclusion drawn from modelling studies. If yes, consider rephrasing it to reflect that, e.g. "Model-based results from these studies seem to suggest that if such thresholds are not crossed, ice sheet retreat can be halted or might even be reversible.", which by the way highlights my observation about repeated info and provides a bridge to the next sentence.

Yes it is a conclusion from the modelling studies. We will change line 56 to: "Model-based studies suggest that, if temperature overshoots are limited, ..."

Lines 57-59: Sounds weird. Consider something like "However, a thorough model-based assessment of the role played by ice sheet-climate feedbacks in the reversibility of enhanced deglaciation rates is currently lacking."

We will include your suggestion, thank you.

Lines 62-67: This paragraph sounds very model-specific for an introduction section, and ignores existing research with other models (e.g. Madsen et al. 2022). It can be easily rephrased to coupled setups in a general sense and acknowledge other groups worldwide working with bidirectional coupling, plus some context. As a bonus, this would solve the issue of introducing the models (and acronyms) twice in the manuscript. CESM-CISM-specific sentences can be moved to the model description below. The remaining 2 mentions of CESM-CISM can be replaced by "a coupled ice sheet and Earth system model" and removed, respectively.

We will consider your suggestion for the revised manuscript and rewrite this paragraph to make it less model-specific.

Line 140: The first half of the simulation design section is just a description of the 1w coupling. I think this could go into the coupling section, which can be divided into 2-w and 1-w for clarity. If the design of 2-w is taken 1:1 from previous studies, I'd like to see it made clear in the text; since at the moment it is somewhat implicit but still unclear. In other words, if any design or simulations are taken directly from previous work, I'd appreciate if there is a clear separation from what is brand new in this study.

Thank you for your suggestion. First of all, to make more clear that the current description in Section 2.2 is about bi-directional coupling, we propose to change line 111 from "By coupling CISM2 with CESM2, ..." to "By applying a 2-way coupling between CISM2 and CESM2, ..." , following a suggestion of referee 2. Then, we will move the 1-way and 2-way design to the end of Section 2.2 as you proposed.
The design of 2-way is taken from Muntjewerf et al. (2020). The 2-way simulation is an extended version of the simulation in Muntjewerf et al. (2020), we will add this in the revised manuscript.

Line 156: kept constant for how long? Would be nice to have the total length of the simulation here as well.

Until we reach year 500, we will add this.

Lines 230-231: Wouldn't it make more sense to compute then the percentage of the continent/island/whatever that experiences ablation?

Yes we did that, we will change the wording "extent of the ablation area" to "percentage of the GrIS that experiences ablation" to make this more clear.

Line 467: 66% more melt than what? 1-way? PI melt? Please specify. Same with other percentages elsewhere.

66% more in 2-way than in 1-way, we will make this more clear and have a check throughout the manuscript for other mentions of percentages.

Line 471: missing "/km" in the units of the rate?

Thank you for pointing out, we will correct the units.

Line 472: what do you mean here? You mean the "real" rate?

Yes, we will indicate that it is the real rate in the revised manuscript.

**References**

Crow, B. R., Tarasov, L., Schulz, M., and Prange, M.: Uncertainties originating from GCM downscaling and bias correction with application to the MIS-11c Greenland Ice Sheet, Clim. Past, 20, 281–296, https://doi.org/10.5194/cp-20-281-2024, 2024.

Gregory, J. M., George, S. E., and Smith, R. S.: Large and irreversible future decline of the Greenland ice sheet, The Cryosphere, 14, 4299–4322, https://doi.org/10.5194/tc-14-4299-2020, 2020.

Madsen, M.S., Yang, S., Aðalgeirsdóttir, G. et al. The role of an interactive Greenland ice sheet in the coupled climate-ice sheet model EC-Earth-PISM. Clim Dyn 59, 1189–1211 (2022). https://doi.org/10.1007/s00382-022-06184-6

Muntjewerf, L., Sellevold, R., Vizcaíno, M., Ernani da Silva, C., Petrini, M., Thayer-Calder, K., Scherrenberg, M. D. W., Bradley, S. L., Katsman, C. A., Fyke, J., Lipscomb, W. H., Lofverstrom, M., and Sacks, W. J.: Accelerated Greenland Ice Sheet Mass Loss Under High Greenhouse Gas Forcing as Simulated by the Coupled CESM2.1-CISM2.1, Journal of Advances in Modeling Earth Systems, 12, e2019MS002 031, https://doi.org/10.1029/2019MS002031, 2020.

---

## Author Comment (AC2)

We thank the reviewers for their constructive feedback on the manuscript. In the following, we provide a short joined response to all reviewers. Thereafter, a response to the specific comments of reviewer 2 is given.

**Joint response to all reviewers**

*On the scope of the manuscript and request to add one-way coupled simulation for the 4x to 1xCO2 reduction scenario* : the scope of the manuscript is to examine Greenland ice sheet and climate **interactions**. Feedbacks are one specific type of these interactions, namely those that involve a bi-directional coupling (initial process is augmented or reduced through the feedback). We will make the interaction-feedback distinction more explicit in the introduction of the reviewed manuscript. Quantification of the albedo feedback for 4xCO2 has been done in previous work (Muntjewerf et al, 2020) by examining the contribution of absorbed solar radiation to the total melt energy and a dedicated simulation is not necessary. For this reason, here we focus on the elevation feedback. Since elevation does not change in the mitigation scenario (mass balance becomes approximately zero), we find that it is unnecessary to explore elevation feedbacks there with a one-way coupled simulation.

In addition, we want to clarify that the primary goal of the manuscript is ***not*** to quantify the difference in melt projections for ice-sheet-only and coupled models. We do this only for our model, and the results will be different for other climate models and surface mass balance schemes. In our paper, this numerical comparison makes one part of the manuscript, with the main focus being the physical ***processes*** of ice sheet and climate interaction, and how our model represents them in the one-way and two-way coupled flavors. We will make this more explicit in the reviewed manuscript.

*Suggestion to run more simulations*: Here we present a set of multi-century "IPCC-type" Earth System Model simulations with a 1 degree atmosphere and dynamical ocean components. This type of model is extremely complex and simulations are computationally very expensive (3,600 core hours are required to run one simulation year). To our knowledge, here we are presenting the first comparison of one-way to two-way simulation with an IPCC-type model. In addition, we present the first assessment of the coupling of global climate, ocean circulation and GrIS snow/firn evolution with an IPCC-type model for a scenario of mitigation. We don't have the means to run more simulations.

*Suggestion to eliminate or move the $CO_2$ reduction simulation to a different paper for consistency or to highlight results separately*: we consider this unnecessary as the common theme here is the assessment of processes of ice sheet and climate interaction. The current structure of this manuscript around the theme of ice sheet-climate interactions first shows the effect of elevation feedbacks by looking at an extreme warming scenario and comparing a set-up with and without evolving GrIS topography, and thereafter addresses other interactions (ocean, snow pack) in the light of a mitigation scenario, aiming to quantify the effect of different interactions and feedback on the GrIS mass balance. Besides, the use of different simulations to address one research question (In our case: "Which interactions between the GrIS and the climate affect the GrIS mass balance?") is not uncommon (e.g., see Gregory et al. (2020), analyzing one 1-way and

several 2-way coupled simulations for different warming scenarios and for multiple mitigation scenarios, around the theme of irreversible mass loss). We propose to make some changes to emphasize more on the common theme in this manuscript (interactions and feedbacks) and the connection between both parts.

To make the common theme clearer we propose to change the title to: "Role of elevation feedbacks and ice-climate interactions on future Greenland melt"

*Request to run more simulations to provide a "one-fits-all" seasonally varying lapse rate for one-way simulations*: we believe this lapse rate will depend both on the modeler choice of climate model forcing and surface mass balance calculation. In this manuscript we do provide a seasonally varying estimate of the temperature lapse rate by comparison of two-way and one-way simulations in CESM. To our knowledge, nobody has provided this sort of estimate. We expect estimates from other models to follow. Crow et al. (2024) is a different type of assessment, where they try different prescribed lapse rates and see which one/type results in a better fit to proxy records.

*Request to clarify one-way simulation design:* the one-way simulation has evolving albedo as this is calculated interactively in the land component. Ice sheet area and elevation are not evolving in the climate components. Meltwater fluxes to the ocean are not evolving. They are prescribed to those calculated in the pre-industrial simulation. We will clarify the simulation design (choices) further in the reviewed manuscript.

*Request to provide justification of fixed lapse rate choice in one-way simulation*: a fixed lapse rate was chosen for consistency with the standard design for sub-grid surface mass balance simulation (downscaling) through elevation classes. Other state-of-the-art downscaling techniques suggested by reviewer 3 are not applicable to an Earth System Model as they are based on high-resolution regional modelling at the scale of 10 km.

*Questions about albedo feedback*: the albedo feedback has been already quantified in a previous study (Muntjewerf et al., 2020). This can be done by looking at the energetic contribution of albedo change (in $W/m^2$) to the total melt energy. That is, there is no need to perform dedicated sensitivity simulations to quantify this feedback. We will make this more explicit in the revised manuscript.

**Response to specific comments of reviewer 2**

Referee comments in black, authors' response in red

Have there been other similar simulations done by other CESM2 users you could include as comparison? Or could this be part of a larger study involving, e.g an overshoot scenario study with CESM2-CISM2?

There have not been run more climate mitigation scenarios with the CESM2-CISM2 set-up. Although our simulation could be of interest for a larger study about overshoot scenarios, our objective is not to only look at the mass balance (and therefore SLR projections), but to look at the processes and interactions involved. Although a larger number of simulations with different mitigation scenarios would be very interesting for SLR projections, at this point it would not add significantly to the story of ice sheet-climate interactions that play an important role in mitigation scenarios, especially compared to the computational costs of our model.

**Response to minor comments**

General comment on figures: some figures might benefit from being slightly wider (e.g. maps). I liked the consistent use of colours but the shades of blue, red and green were sometimes difficult to tell apart, especially in figure 2d and again in figures 7b and 9b. Could you keep the same main colours but change the tint/shade (i.e. make darker tones even darker and lighter ones lighter)?

Thanks for bringing this to our attention, we will change the colors in fig 2d, 7b and 9b and have a look at the other figures as well. We will make fig 1, 3, 8 and 10 wider.

**Abstract**
p1, line 10: I suppose the lapse rate you're mentioning is the lapse rate used to downscale temperature from the elevation tiles to the ice sheet grid. It would be useful to add that information.

Yes, we will add that.

p1, lines 12-13: add that it is for the 2-way coupled simulation

We will change this to: "Furthermore, we analyze a simulation branched in year 350 from our 2-way coupled simulation in which we annually reduce atmospheric CO2 by 5% until PI concentrations are reached."

**1. Introduction**
p2, line 52: what do you mean by "such a period"? A certain length of time or simply that it's a warmer period?

A period in which a certain temperature threshold is surpassed. We will change this part to: "The rapidly increasing global temperatures call for the investigation of 'overshoot'

scenarios, where this temperature threshold is surpassed. Such an overshoot could have large implications for the evolution of the GrIS-induced SLR, as GMSL could rise substantially under the larger temperatures during an overshoot period." (after following some suggestions from referee 1 as well).

**2. Method**

p4, lines 96-97: I'm not sure I understand these 2 sentences. Is the snowpack thickness reset to 10m at the beginning of every year? Meaning that if the thickness at the end of the year is 10+X m, X m is the positive SMB that is transferred to CISM2? And, in the second sentence, what do you mean by "further melt"?

The maximum snow thickness is 10 m. When the snowpack exceeds 10m (10 + X), then X m will be transferred to CISM as positive SMB to increase ice thickness (ice accumulation).
We will change "further melt" to "further melt of ice", as this concerns the melt after the 10 m w.e. has already melted (ice ablation).
As the snowpack is only part of CLM, changes that only occur in the snowpack (meaning no ice ablation/accumulation), will not be communicated to CISM.

p4, line 119: I don't think the definition of refreezing capacity is necessary here, as you only mention it much later in the manuscript (p19, end of section 5). Instead, I would just change line 376 (p19) in "The refreezing capacity (amount of refreezing divided by the amount of available water) peaks earlier..."

Thank you, we will follow your suggestion.

p5, line 126 (then p6, line 146 and p11, line 252): The way you wrote these different sentences, I am not sure whether the fixed lapse rate of -6K/km is used in both the 1- and 2-way simulations. From line 126, I think yes but then I was a bit puzzled when reading "as is done in the 2-way coupled configuration" in line 252. Finally, in lines 250 to 254, you mention a computed lapse rate that you compare to the fixed lapse rate of the 1-way simulation.

Yes the fixed lapse rate is used in both the 1-way and 2-way set-up to interpolate from CLM to CISM, using elevation classes. In the 2-way set-up, this lapse rate is only used to interpolate from the coarser CLM grid to the finer CISM grid and allows for taking the nonuniform topography within the larger CLM grid cells into account. In 1-way, the lapse rate is used to describe elevation changes due to melt as well, as the CLM topography is fixed. We will change line 251-252 from "... we compute lapse rates..." to "... we compute the lapse rates resulting from elevation change..." to make this more clear.

If I understood correctly, the lapse rate used for downscaling the temperature from CESM's to CISM's grid is fixed in both simulations. I'd add in line 126 that it is the case in both simulations, to make it clear the first time it's mentioned. And I'd remove "as it is done in the 2-way coupled configuration" entirely in line 126.

We propose to change line 111 from "By coupling CISM2 with CESM2, ..." to "By applying a 2-way coupling between CISM2 and CESM2, ..." to point out that the coupling description in section 2.2 is about bidirectional coupling. We will remove as it is done in the 2-way coupled configuration".

Then, in section 4, lines 250-254 I think the lapse rates that you mention you're computing are computed offline, in the same way as you computed the lapse rates for the ME feedback. I'd add here for clarity that those are computed offline I would then add, either in line 252 or 253, something along the line of "compared them with the fixed applied lapse rate used to downscale temperature and LW during the simulations". I'd also remove the 1-way simulation mention if the 2-way simulation also uses a fixed lapse rate for temperature and LW. I think adding that would make the reader know immediately which lapse rates you are referring too and would make the reading easier.

The lapse rates computed are the 'real' changes of temperature and LW when using 2-way coupling and are computed by comparing the changes in temperature and LW fields with the changes in elevation.

p5, line 137: You're mentioning the fact that, since the coupling from POP2 to CISM2 is not implemented yet (presumably because you need a way to downscale ocean temperatures onto the ice sheet grid in order to be able to resolve the fjords), there is no direct influence of the ocean on the ice sheet via it's forcings on marine terminating glaciers. Could you add a few words about the potential biases this could lead to and the processes involved?

As increases in sea surface temperature are not communicated back, we will not see increases in ice discharge resulting from this. We will add a bit on that: "Therefore ocean-forced melting of marine-terminating glaciers is not accounted for, which could lead to biases in the computed ice discharge." This is however not of great importance for the simulations done in this study, as ice discharge will go to zero when the ice sheet becomes land-terminating under an extreme warming scenario.

p6, line 165 to p7, line 193: Section 2.4 (metrics definitions) I don't think this section is necessary in this form. Some of the concepts are useful to define in a thesis but are well known to the readers of scientific papers (e.g. the definition of ELA) and others are only used much later in the manuscript and should, in my opinion be moved there. I'd keep the definitions of lapse rates, GBI and the moving average and remove the ELA, NAO and IVT.
- Lapse rates: I am not entirely sure what the lapse rates refer to here. I think you're using them to isolate the melt-elevation feedback as this would be the only way to evaluate that the ME feedback leads to more melt in the 2-way simulation since the SMB doesn't decrease as much in the 2-way simulation and the snowfall doesn't differ much. I'd move the definition to section 4.1.
- GBI: I'd keep the definition in the manuscript as you're using the modified GBI proposed by Hanna et al. (2018), which they call GB2. I'd move the definition to section 4.2 and would add that what you're using is called GB2 in Hanna et al.

- Moving averages: I'd keep that one to remind the reader that the length of the moving average changes during the simulations but I'd put it at the end of section 2.3 (simulation design).
-

Thanks for the suggestion, we intent to make your proposed changes in the revised manuscript.

**3. Simulated mass loss**
p8, lines 216-218: Is there a numerical threshold in the NAMOC index for you to consider it to collapse or are you looking at increased rates of change?

We look at increased rates of change.

**4. Climate feedbacks**
p12, line 262: I'd add orange dashed line after Figure 5c so the reader can spot the point without reading the caption.

Good suggestion, we will add that.

p11, caption figure 4: the blue line shows the monthly mean surface temperature, not the red line.

Thanks for spotting this mistake, we will change it.

**6. Discussion**
p21, line 425: By considering the surface temperature ? What does considering mean here?

The lapse rates are strongly influenced by surface temperature, as for a melting surface, much of the available energy will be used to melt the surface instead of heating the atmosphere, leading to smaller lapse rates. We will change this line to "...by considering whether the surface temperature has reached melting point." to make this more clear.

**Typos, spelling, punctuation**
p4, line 103: ice thermodynamicS model instead of thermodynamic? Like in ice dynamics model?
p4, line 106: same p6, line 140: simulationS design?

p6, line 146: -6 K km-1 instead of K/km (as p5, line 126)
p6, line 148: "In contrast, the 2-way coupled run..". to the 1-way coupled run in not necessary here as you were just talking about the 1-way coupled run in the previous sentence.
p6, line 157: I'm not sure the last part of the sentence (after which) is grammatically correct. I would write "the 4xCO2 scenario is an extreme warming scenario and, after the year 140, has a similar radiative forcing to that of the SSP5-85 scenario at the end of the 21st century".

p7, lines 197&203+p8, line 206: 20-year centered moving average (with a hyphen)

p8, line 207: returns within one standard deviation OF the PI mean.

p10, line 244: no comma after we account for this
p10, line 245: comma after while if you put one after 1-way simulation

p12, line 264: no comma after simulation

p15, line 311: not significantly instead of not significant

p16, line 335: here you express SLR in cm but in line 321 you express it in m. Can you check throughout the manuscript and pick one?

p18, line 372: no comma between forcing and are

p21, line 395: as opposed to?
p21, line 399: comma before although
p21, line 403: comma after cloud cover
p21, line 404: the sentence is a bit long so I'd start a new one after transmissivity (This aligns...)

p22, line 433: tens of thousands of years
p22, line 448: no comma after level
p22, line 449: no comma after small
p22, line 450: if we were extending I think

Many thanks for reading the manuscript so thoroughly, we will incorporate you proposed changes regarding spelling, typos and punctuation.

**References**

Crow, B. R., Tarasov, L., Schulz, M., and Prange, M.: Uncertainties originating from GCM downscaling and bias correction with application to the MIS-11c Greenland Ice Sheet, Clim. Past, 20, 281–296, https://doi.org/10.5194/cp-20-281-2024, 2024.

Gregory, J. M., George, S. E., and Smith, R. S.: Large and irreversible future decline of the Greenland ice sheet, The Cryosphere, 14, 4299–4322, https://doi.org/10.5194/tc-14-4299-2020, 2020.

Muntjewerf, L., Sellevold, R., Vizcaíno, M., Ernani da Silva, C., Petrini, M., Thayer-Calder, K., Scherrenberg, M. D. W., Bradley, S. L., Katsman, C. A., Fyke, J., Lipscomb, W. H., Lofverstrom, M., and Sacks, W. J.: Accelerated Greenland Ice Sheet Mass Loss Under High Greenhouse Gas Forcing as Simulated by the Coupled CESM2.1-CISM2.1, Journal of Advances in Modeling Earth Systems, 12, e2019MS002 031, https://doi.org/10.1029/2019MS002031, 2020.

---

## Author Comment (AC3)

We thank the reviewers for their constructive feedback on the manuscript. In the following, we provide a short joined response to all reviewers. Thereafter, a response to the specific comments of reviewer 3 is given.

**Joint response to all reviewers**

*On the scope of the manuscript and request to add one-way coupled simulation for the 4x to 1xCO2 reduction scenario* : the scope of the manuscript is to examine Greenland ice sheet and climate **interactions**. Feedbacks are one specific type of these interactions, namely those that involve a bi-directional coupling (initial process is augmented or reduced through the feedback). We will make the interaction-feedback distinction more explicit in the introduction of the reviewed manuscript. Quantification of the albedo feedback for 4xCO2 has been done in previous work (Muntjewerf et al, 2020) by examining the contribution of absorbed solar radiation to the total melt energy and a dedicated simulation is not necessary. For this reason, here we focus on the elevation feedback. Since elevation does not change in the mitigation scenario (mass balance becomes approximately zero), we find that it is unnecessary to explore elevation feedbacks there with a one-way coupled simulation.

In addition, we want to clarify that the primary goal of the manuscript is ***not*** to quantify the difference in melt projections for ice-sheet-only and coupled models. We do this only for our model, and the results will be different for other climate models and surface mass balance schemes. In our paper, this numerical comparison makes one part of the manuscript, with the main focus being the physical ***processes*** of ice sheet and climate interaction, and how our model represents them in the one-way and two-way coupled flavors. We will make this more explicit in the reviewed manuscript.

*Suggestion to run more simulations*: Here we present a set of multi-century "IPCC-type" Earth System Model simulations with a 1 degree atmosphere and dynamical ocean components. This type of model is extremely complex and simulations are computationally very expensive (3,600 core hours are required to run one simulation year). To our knowledge, here we are presenting the first comparison of one-way to two-way simulation with an IPCC-type model. In addition, we present the first assessment of the coupling of global climate, ocean circulation and GrIS snow/firn evolution with an IPCC-type model for a scenario of mitigation.  We don't have the means to run more simulations.

*Suggestion to eliminate or move the $CO_2$ reduction simulation to a different paper for consistency or to highlight results separately*: we consider this unnecessary as the common theme here is the assessment of processes of ice sheet and climate interaction. The current structure of this manuscript around the theme of ice sheet-climate interactions first shows the effect of elevation feedbacks by looking at an extreme warming scenario and comparing a set-up with and without evolving GrIS topography, and thereafter addresses other interactions (ocean, snow pack) in the light of a mitigation scenario, aiming to quantify the effect of different interactions and feedback on the GrIS mass balance. Besides, the use of different simulations to address one research question (In our case: "Which interactions between the GrIS and the climate affect the GrIS mass balance?") is not uncommon (e.g., see Gregory et al. (2020), analyzing one 1-way and

several 2-way coupled simulations for different warming scenarios and for multiple mitigation scenarios, around the theme of irreversible mass loss). We propose to make some changes to emphasize more on the common theme in this manuscript (interactions and feedbacks) and the connection between both parts.

To make the common theme clearer we propose to change the title to: "Role of elevation feedbacks and ice-climate interactions on future Greenland melt"

*Request to run more simulations to provide a "one-fits-all" seasonally varying lapse rate for one-way simulations*: we believe this lapse rate will depend both on the modeler choice of climate model forcing and surface mass balance calculation. In this manuscript we do provide a seasonally varying estimate of the temperature lapse rate by comparison of two-way and one-way simulations in CESM. To our knowledge, nobody has provided this sort of estimate. We expect estimates from other models to follow. Crow et al. (2024) is a different type of assessment, where they try different prescribed lapse rates and see which one/type results in a better fit to proxy records.

*Request to clarify one-way simulation design:* the one-way simulation has evolving albedo as this is calculated interactively in the land component. Ice sheet area and elevation are not evolving in the climate components. Meltwater fluxes to the ocean are not evolving. They are prescribed to those calculated in the pre-industrial simulation. We will clarify the simulation design (choices) further in the reviewed manuscript.

*Request to provide justification of fixed lapse rate choice in one-way simulation*: a fixed lapse rate was chosen for consistency with the standard design for sub-grid surface mass balance simulation (downscaling) through elevation classes. Other state-of-the-art downscaling techniques suggested by reviewer 3 are not applicable to an Earth System Model as they are based on high-resolution regional modelling at the scale of 10 km.

*Questions about albedo feedback*: the albedo feedback has been already quantified in a previous study (Muntjewerf et al., 2020). This can be done by looking at the energetic contribution of albedo change (in $W/m^2$) to the total melt energy. That is, there is no need to perform dedicated sensitivity simulations to quantify this feedback. We will make this more explicit in the revised manuscript.

**Response to specific comments of reviewer 3**

**Abstract**
- P1. L6-8: Please add a time reference for your simulations and specify the level of CO2 at which you start.

We will add that these are multi-century simulations and that they start from PI CO2 conditions.

- P1, L10-12: "We also find that a uniform temperature lapse rate misrepresents temperature changes in the ablation area, leading to an overestimation of the positive melt-elevation feedback in the 1-way coupled simulation, resulting in an overestimation of mass loss." I think it's inappropriate to consider that the overestimation of the melt-elevation feedback by the 1-way simulation is an original and new result of your study as you already mention in your introduction (P2 L41-43) and in your discussion that other different studies with similar one-way experiments have already the same conclusion. You should rephrase and nuance this sentence.

You are correct that the overestimation in 1-way is not a new result of our study. However, the novelty here is the attribution of the physical processes causing this overestimation. To make this more clear, we will change this line to: "We also attribute part of the overestimation of mass loss in the 1-way coupled simulation to an overestimation of melt in the ablation area, caused by the use of a uniform temperature lapse rate."

- P1 L12-13: precise that you extend your 2-way coupled simulation instead of a new experiment.

We will change this line to "Furthermore, we analyze a simulation branched in year 350 from our 2-way coupled simulation in which we annually reduce atmospheric CO2 by 5% until PI concentrations are reached."

**Method**
- P4. L94-99 "The model has a fixed number of vertical layers for the soil, whereas there is a variable number of layers for snow and firn, with a maximum snow depth of 10 m water equivalent (w.e.). The model allows for compaction of snow into firn. Accumulation of snow over 10 m w.e. in a grid cell is transferred as positive SMB to CISM2. If snow and firn are melted away, further melt is transferred as negative SMB (ice ablation) to CISM2."Does it mean that having less than 10 m w.e. of accumulation is not considered in the SMB ? Same for ablation?

In this coupled set-up, the snow layer is only part of CLM, meaning that changes in only the snowpack will not be communicated to CISM. Only ice ablation (ablation occurring when the snow and firn layer in CLM have completely disappeared) and compaction of snow and firn into ice (when the snow layer exceeds 10 m w.e.) are communicated to CISM.

- P4, L119: Please add equations for SMB and refreeze capacity to illustrate and summarize how SMB and refreeze are considered.

We will add the equations for SMB and SEB here. We will leave the refreezing capacity out in this part and add it in Section 5, where refreezing is discussed (as suggested by reviewer 2).

- P4: In general, has your model and specifically the representation of the SMB, the key feature of your coupling experiment already been evaluated against observation? How robust are your results? What are the range of bias for the different processes you are representing?

As evaluated by van Noël et al. (2020) and Kampenhout et al. (2020), CESM2 yields a realistic SMB, compared to in-situ observational data and RACMO model data. Since the GrIS topography we get after the spin-up procedure is slightly different than the present-day topography, the SMB will be biased to this as well. As for ice discharge, there is no increased ocean forcing from increasing sea temperatures, since coupling to the ocean modelling is uni-directional, which introduces a bias as well. This is however not of great importance for the simulations done in this study, as ice discharge will go to zero when the ice sheet becomes land-terminating under an extreme warming scenario.

- P5 L126 & L.145: "The resulting climate and SMB are downscaled using elevation classes (Sellevold et al., 2019), using a temperature lapse rate of -6 K/km and are interpolated onto the CISM grid". The way you explain how the temperature lapse rate is used to interpolated the SMB is not clear enough for me. Are the SEB and SMB also downscaled with a lapse rate? Or are they calculated on the CISM grid after the temperature has been interpolated using the constant lapse rate? This part of your method should clarify as this is the only way you consider the melt-elevation feedback in your 1-way coupling experiment. Furthermore, could you briefly explain the elevation class method?

For every grid cell in CLM, a set of atmospheric variables (a.o. temperature), are computed for 10 different elevations by means of a lapse rate. Then, using these atmospheric variables, the SMB and SEB for all elevation classes is computed. The CLM output is then the area-weighted average of these elevation classes. The CISM SMB is then computed by interpolation using the elevation of the grid cell in CISM. More on this method can be found in Sellevold et al. (2019), but we will also add a brief explanation to the manuscript.

- P6, L142-143: How the initial topography of the ice sheet is retrieved? Is it the usual topography used by CAM6? Or is there a step of initialization for CISM? Same questions for freshwater fluxes, from where are they come from? Also, if you use a different topography than the usual one for CAM6, how the atmospheric module is answering to, sometimes, big differences in elevations?

The topography of the 1-way and 2-way simulation are the same in year 0, this topography is the one computed from the model spin-up as described by Lofverstrom et al. (2020),

and is slightly different from the present-day topography. Initial freshwater fluxes are computed during the spin-up procedure as well.
The CAM, CLM and CISM topographies are connected in a 2-way coupled set-up. Using the downscaled SMB, every model year a new CISM topography is computed. Then, a new CLM topography (on a coarser grid) will be computed from the CISM topography for every model year as well. Besides, CAM receives an updated topography from CLM every 5 or 10 years (simulation dependent).

- P6 L.152-153: "In contrast to the surface topography, the surface albedo is update  as a response to ice sheet melting in both the 1-way and 2-way coupled simulations." I had to read several times this explanation to really understand what it means. I guess then this is not the best way to explain it. You should more deeply explain what implies this consideration of the albedo. Also, as in the 2-way simulation the ice sheet is retreating, is the albedo-feedback not mitigated by the smaller area of higher albedo (ablation area)?

Thank you for pointing this out, we will change this to: "In contrast to the surface topography, the surface albedo is updated in both the 1-way and 2-way coupling. The model allows for the surface albedo to change. Therefore, as snow and ice melt or snow accumulates, the albedo will be updated."
Regarding the mitigation of the albedo-feedback, yes this is the case, however only later (starting around year 450), this plays a role. Looking at fig 2h, you can see the percentage of area that is ablation area stabilize, which is due to the melting of the margins.

- P6 L.155: Please specify here how much time your experiments are running. For more clarity, I suggest also to refer to figures 2a and 9a to illustrate you experiment designs.

Yes, we will add that, and we will follow your suggestion to refer to the figures.

- P6 L164: Concerning your control simulation, why only have one control simulation with 2-way coupling method? Did you compare it with a similar experiment but with the 1-way method? In other words, does the method to represent the melt-elevation feedback influence the control simulation even if the melt-elevation feedback is supposed to be weak with this CO2 level?

No, we did not do a 1-way control simulation. In lines 200-202, we explain a bit about the choice for this. As in the control simulation the mass balance is nearly zero (0.03 mm/yr), the melt-elevation feedback will not play a role, like you said. As the only difference is that the surface topography in CLM and CAM can evolve when using 2-way coupling, we expect a very similar control run when done with 1-way coupling. Considering how computationally expensive our model is, we do not see enough added value in a 1-way coupled control simulation, and therefore only use the 2-way coupled control simulation.

- P7 L179-180 and 183: Could you specify how you normalize with respect to the control simulation? Is it mean you're considering the variability of the GBI/NAO from the control run to normalize the GBI/NAO from 1w- and 2w-experiment?

Yes, that is correct, we use the variability from the control run for the normalization. We will change lines 179-180 from "normalize with respect to the control simulation" to "normalize using the variability of the control simulation". The same holds for line 182.

- P7 L190-193: Usually, a period of 30 years is considered to talk about climatology mean. Is the choice of 20 years/data point influence your results compared to a 30- year average?

Here, we chose to stick to the 20 years used in earlier studies of CESM2-CISM2 (e.g. Muntjewerf et al. (2020)). As changes in the $CO_2$ ramp-up and during the $CO_2$ ramp-down are relatively fast, a 20-year mean might capture strong trends better. However, we do not expect a that using a 30-year mean would influence our results strongly.

**Results**
- Fig 3c: The difference between a) and b) gives still a rate per year. This is not sound to the topography feedback to me. I suggest also illustrating the final result in terms of topography with the differences in meters between topography of 1-way and 2-way for year 500.

Thanks for mentioning this, you are correct. Fig 3c shows the melt differences between 1-way and 2-way (which are then caused by different elevation feedbacks), but this is not the same as the elevation feedback. We will change the title of Fig 3c to "Difference" to make sure there is no confusion about this.
For the topography differences we would like to refer to Fig A1a-e (ice thickness maps from CISM), we will add a reference to this in the text around Fig 3.

- P11, L251: Please justify why you also look at the LW down (and not LWup, SWdown, or up).

The downscaling of LW_down using elevation classes has a large influence on the uncertainty of the SEB (Sellevold et al., 2019). As LW is also dependent on temperature, it would be interesting to see how the lapse rates of the LW and near surface air temperature relate. As for the upward LW component, this will be largely surface temperature dependent, meaning that attributing changes in LW_up to elevation change is difficult.

- P12 L 273-275: I suggest at least adding these references to describe atmospheric blocking: Hanna et al. 2014, McLeod and Mote 2016.

Thank you for the suggestion, we will add these references.

- P12 L276: Are the differences in GBI between both simulations not simply linked to change in height of the surface, at least partly, thus decreasing the geopotential height of 500hPa? If your surface is lowering, the height of the 500hPa geopotential is also lowering, especially as you have differences in surface elevation up to 1000m after 500 years of coupling. And then this GBI decrease will not be entirely due to "real" changes in blocking event regime, and more generally changes in larger scale circulation. Also, I'm surprise to have such differences, even just in winter, in GBI, and not correlated at all with differences

in NAO, as these 2 indexes are partly anticorrelated for the current period (Hanna et al., 2015).

Yes, they are related to height changes. As you say, the 500 hPa geopotential will indeed lower when the surface height is lowering. The way we see it, the lowering of such a big atmospheric 'obstacle' as the GrIS, will result in less persistent high-pressure fields, as these are able to travel more easily. Mullen,1989; Narinesingh et al., 2020; Sellevold et al., 2022 found a positive relationship between orography and blocking events, indicating that a significantly lowering topography can change the blocking regime. In our view, the cause of the changes in blocking is the lowering of the topography, and therefore correcting for the changes in topography would not make that much sense to us. We will however nuance our results a bit more, as it is possible that, as you say, we do not only tackle the changes in blocking but in other circulation patterns as well. We will change the many uses of the term "atmospheric blocking" to "Greenland blocking index". We will add to line 277: "As the large changes in GrIS topography might have large influences on the atmospheric circulation around Greenland as a whole, it is possible that our computed changes in blocking index do not only consist of changes in blocking but are influenced by other changes in atmospheric circulation as well."

- P13 L 281-292: In this part of your results, you should consider the influence of altitude on the GBI computation (as explained in the former comment) before reaching any conclusions on the relationship between melting and blocking events. Same comment for the discussion (P21 L 406-415) even if this part is already well nuanced.

For the part about the results section, we would like to refer to the answer above. Regarding the discussion, here we would like to change the term "blocking" to "blocking index" as well. Besides, we would like to change lines 410-411 to: "However, further investigation into the causes of these changes in blocking and their relationship with melt is necessary to make a definite attribution, especially since the robustness of the blocking index towards large topographic changes has not been evaluated."

- P 16 L 315-317: Despite the accuracy of the explanation concerning precipitation, this analysis could be mitigated by comparing the relative importance of the feedbacks mentioned, compared with melt-elevation feedback (Fig 3 and Fig A1h VS. Fig 8) as precipitation is a much lower contributor to the differences between 1- and 2-way experiments.

Here, we would like to refer to line 400 of the discussion and Fig A3, stating that the contribution of the larger snowfall in 2-way is limited. We will add "limited (..), *compared to other feedbacks.*" to this line.

- P16 L345: "However, the retreated ice sheet margins result in a smaller contribution of ice discharge to the mass balance (Figure 9d)." If I'm right, could you specify that you compared to the PI situation in this sentence? Also, when you consider the "first" state of your comparison (Table 2), please, indicate the years, to be clearer.

Thank you for pointing out that this is unclear, we will indicate that this is compared to PI and add the years in the comparison.

- In this comparison (Table 2), you should insist on the fact that, by recovering the same global temperature anomaly, the state of the ice sheet is quite different, as well as the components engaged in the total mass balance. It could be also interesting for your analysis to have a spatial representation of the ice sheet extent for these 2 specific states, and more generally to illustrate what becomes the ice sheet after such a decrease in CO2.

For us, the point of showing this comparison, is to make clear that, despite the similar degree of warming, the ice sheet is in a completely different state. We will make this more clear. We agree that a spatial representation of these two states would definitely be interesting to add, therefore we intent to add this to the revised manuscript.

**Discussion**
- P21 L412 : "However, Hanna et al. (2018)", Add Delhasse et al 2021, which is the updated version of Hanna et al. 2018 with CMIP6 models.

Thank you for this suggestion, we will add this reference.

- P22 L432: Please specify that you're mentioning 1.1m of SLR contribution.

Thanks for spotting this mistake.

**Typo**
- P2 L51: Please define GMSL;

We will define it in the revised manuscript.

- L672: The reference of Sellevold et al. 2019 should be updated as the paper is not in discussion anymore.

Thank you for finding this mistake, we will update this reference.

**References**

Crow, B. R., Tarasov, L., Schulz, M., and Prange, M.: Uncertainties originating from GCM downscaling and bias correction with application to the MIS-11c Greenland Ice Sheet, Clim. Past, 20, 281–296, https://doi.org/10.5194/cp-20-281-2024, 2024.

Delhasse A, Hanna E, Kittel C, Fettweis X. Brief communication: CMIP6 does not suggest any atmospheric blocking increase in summer over Greenland by 2100. Int J Climatol. 2021;1–8. https://doi.org/10.1002/ joc.6977

Gregory, J. M., George, S. E., and Smith, R. S.: Large and irreversible future decline of the Greenland ice sheet, The Cryosphere, 14, 4299–4322, https://doi.org/10.5194/tc-14-4299-2020, 2020.

Hanna, E., Fettweis, X., Mernild, S.H., Cappelen, J., Ribergaard, M. H., Shuman, C.A., Steffen, K., Wood, L. and Mote, T.L. (2014) Atmospheric and oceanic climate forcing of the exceptional Greenland ice sheet surface melt in summer 2012. International Journal of Climatology, 34, 1022–1037. https://doi.org/10.1002/ joc.3743.

Hanna, E., Fettweis, X., and Hall, R. J.: Brief communication: Recent changes in summer Greenland blocking captured by none of the CMIP5 models, The Cryosphere, 12, 3287–3292, https://doi.org/10.5194/tc-12-3287-2018, 2018.

Hanna, E., Cropper, T.E., Jones, P.D., Scaife, A.A. and Allan, R. (2015) Recent seasonal asymmetric changes in the NAO (a marked summer decline and increased winter variability) and associated changes in the AO and Greenland blocking index. International Journal of Climatology, 35, 2540–2554. https://doi.org/10.1002/joc.4157.

Lofverstrom, M., Fyke, J. G., Thayer-Calder, K., Muntjewerf, L., Vizcaino, M., Sacks, W. J., Lipscomb, W. H., Otto-Bliesner, B. L., & Bradley, S. L. (2020). An Efficient Ice Sheet/Earth System Model Spin-up Procedure for CESM2-CISM2: Description, Evaluation, and Broader Applicability. Journal of Advances in Modeling Earth Systems, 12(8), e2019MS001984. https://doi.org/10.1029/2019MS001984

McLeod, J.T. and Mote, T.L. (2016) Linking interannual variability in extreme Greenland blocking episodes to the recent increase in summer melting across the Greenland ice sheet. International Journal of Climatology, 36, 1484–1499. https://doi.org/10.1002/joc.4440.

Mullen, S. L.: The Impact of Orography on Blocking Frequency in a General Circulation Model, Journal of Climate, 2, 1554–1560, https://doi.org/10.1175/15200442(1989)002<1554:TIOOOB>2.0.CO;2, 1989.

Muntjewerf, L., Sellevold, R., Vizcaíno, M., Ernani da Silva, C., Petrini, M., Thayer-Calder, K., Scherrenberg, M. D. W., Bradley, S. L., Katsman, C. A., Fyke, J., Lipscomb, W. H., Lofverstrom, M., and Sacks, W. J.: Accelerated Greenland Ice Sheet Mass Loss Under High Greenhouse Gas Forcing as Simulated by the Coupled CESM2.1-CISM2.1, Journal

of Advances in Modeling Earth Systems, 12, e2019MS002 031, https://doi.org/10.1029/2019MS002031, 2020.

Narinesingh, V., Booth, J. F., Clark, S. K., and Ming, Y.: Atmospheric Blocking: The Impact of Topography in an Idealized General Circulation Model, Weather and Climate Dynamics Discussions, https://doi.org/10.5194/wcd-2020-2, 2020.

Noël, B., Van Kampenhout, L., Van De Berg, W. J., Lenaerts, J., Wouters, B., & Van Den Broeke, M. R. (2020). Brief communication: CESM2 climate forcing (1950–2014) yields realistic Greenland ice sheet surface mass balance. The Cryosphere, 14(4), 1425-1435.

Sellevold, R., van Kampenhout, L., Lenaerts, J. T. M., Noël, B., Lipscomb, W. H., and Vizcaino, M.: Surface mass balance downscaling through elevation classes in an Earth system model: application to the Greenland ice sheet, The Cryosphere, 13, 3193–3208, https://doi.org/10.5194/tc-13-3193-2019, 2019.

Sellevold, R., Lenaerts, J. T. M., and Vizcaíno, M.: Influence of Arctic sea-ice loss on the Greenland ice sheet climate, Climate Dynamics, 58, 179–193, https://doi.org/10.1007/s00382-021-05897-4, 2022.

van Kampenhout, L., Lenaerts, J. T., Lipscomb, W. H., Lhermitte, S., Noël, B., Vizcaíno, M., ... & van den Broeke, M. R. (2020). Present-day Greenland ice sheet climate and surface mass balance in CESM2. *Journal of Geophysical Research: Earth Surface*, *125*(2), e2019JF005318.

---

## Referee Report (RR1)

A major remark common to all 3 reviewers was about the overall scope of the manuscript. The reviewers felt that either a simulation reintroducing PI conditions was not in the scope of the manuscript or that it should have an equivalent 1-way simulation to follow a pattern similar to the comparison between the 2 4CO2 simulations. The authors have justified their reasons for not running more simulations and made changes throughout the manuscript (including the title) to better reflect the common theme between the two parts.

The authors have also responded to the reviewers' minor comments and incorporated them in the revised manuscript. They have also paid particular attention to specific remarks about the wording and sentences connections to improve the clarity of the manuscript.

I only have a few minors comments but I suggest that the authors simply take a look a them before sending the manuscript to copy editing as these don't require another round of review.

**Minor comments**

p1 - L10-12: We also attribute part of the overestimation of mass loss in the 1-way coupled simulation to an overestimation of melt in the ablation area, caused by the use of a uniform temperature lapse rate ***to describe elevation changes.***

The last part of the sentence was added as a response to reviewer 2 requesting to specify that the temperature lapse rate was the one used to downscale temperature from the atmospheric grid to the ice sheet grid. It is true that it reflects elevation changes but the way you wrote it sounds a bit weird.

I suggest to change it to ***caused by the use of a uniform temperature lapse rate used to reflect the elevation differences between the atmospheric and ice sheet grids.***

p2 - L50: I think ***in which*** *this temperature threshold is surpassed* is more grammatically correct than where. I'm not entirely sure so please double check

p2 - L52: ***higher*** temperatures instead of ***larger***

p5 - L131-132: as you've introduced a sentence explaining the SMB is computed on elevation classes just before, I would replace ***the sum of the downscaled SMB*** by something along the lines of ***the sum of the SMB downscaled from the elevation classes*** so readers that are unfamiliar with the use of elevation classes models can make the link with line 125

p5 - L145: communicated ***to***

p 22 - L 416: Blocking projections should ***therefore*** be treated with caution